# Dynamic modulation of modal coupling in microelectromechanical gyroscopic ring resonators

Xin Zhou[1,2], Chun Zhao[1], Dingbang Xiao[2]*, Jiangkun Sun[1,2], Guillermo Sobreviela[1], Dustin D. Gerrard[3], Yunhan Chen[3], Ian Flader[3], Thomas W. Kenny[3], Xuezhong Wu[2] & Ashwin A. Seshia [1]*

Understanding and controlling modal coupling in micro/nanomechanical devices is integral to the design of high-accuracy timing references and inertial sensors. However, insight into specific physical mechanisms underlying modal coupling, and the ability to tune such interactions is limited. Here, we demonstrate that tuneable mode coupling can be achieved in capacitive microelectromechanical devices with dynamic electrostatic fields enabling strong coupling between otherwise uncoupled modes. A vacuum-sealed microelectromechanical silicon ring resonator is employed in this work, with relevance to the gyroscopic lateral modes of vibration. It is shown that a parametric pumping scheme can be implemented through capacitive electrodes surrounding the device that allows for the mode coupling strength to be dynamically tuned, as well as allowing greater flexibility in the control of the coupling stiffness. Electrostatic pump based sideband coupling is demonstrated, and compared to conventional strain-mediated sideband operations. Electrostatic coupling is shown to be very efficient, enabling strong, tunable dynamical coupling.

[1] Nanoscience Centre, University of Cambridge, Cambridge CB3 0FF, UK. [2] Department of Intelligent Machinery and Instruments, College of Intelligence Science, National University of Defense Technology, Changsha 410073, China. [3] Department of Mechanical Engineering, Stanford University, Stanford, CA 94305, USA. *email: dingbangxiao@nudt.edu.cn; aas41@cam.ac.uk

The interactions of micro- and nanomechanical resonators with various physical fields have been researched for several decades and have been engineered into specific device formats for applications to timing and frequency control[1,2], sensing[3–6], as well as more fundamental studies in alternative approaches to information processing[7–9], and quantum science[10–13]. Several recent studies have also focused on fundamental emergent behaviors in coupled micro/nanomechanical systems[14–17] and the mutual coupling between two distinct mechanical resonators or modes[18–27]. By regarding one of the coupled mechanical modes as a phonon cavity, optomechanics-like dynamical operations, such as cooling (dynamical coupling)[22,28–31], amplification[22,31,32], squeezing[33,34], coherent manipulation[28,35,36], and phonon lasing[30,37], have been demonstrated in micro- and nanomechanical systems. Mode coupling mechanisms based on physical mechanical linkages[28], dielectric coupling[21,35], tension-induced parametric coupling[18,23,29], and internal resonance[20,25] have been previously investigated. However, physical insight is often limited due to the limited experimental control of mode coupling in such systems. Often, manufacturing tolerances or small asymmetries can have a huge impact, and the ability to define modal coupling parameters by design is still limited, often even elusive. Static electrostatic tuning in such systems is possible but provides for only weak tunability, and the ability to define and exercise good control on modal coupling in micro- and nanomechanical resonators remains a key challenge limiting practical applications as well.

In this paper, we demonstrate that dynamic capacitive tuning can enable significant tunability of modal coupling in microelectromechanical devices. The intrinsic modal coupling in such systems can be tuned by the nonlinear fields associated with parallel plate transducers. Such fields can be dynamically modulated enabling a further knob on the tunability of system response. A vacuum-sealed micromachined gyroscopic ring resonator is employed as the experimental testbed in this work. Such a system demonstrates linear hybrid coupling between the gyroscopic near-degenerate modes due to configurational/structural asymmetries arising from manufacturing tolerances or material properties, and misalignment between the principal axis and the detection electrode. We demonstrate that, akin to the tension-induced stiffness hardening mechanisms[18,22,23], electrostatically induced stiffness softening can also provide a nonlinear parametric coupling term. While the static linear and nonlinear interactions generated by the electrostatic field have been previously studied[38,39], capacitive nonlinear parametric coupling between normal modes in a single resonator has not been presented previously. Here, it is shown that apart from the tension-induced parametric coupling, this electrostatic nonlinear parametric coupling may reside, or even dominate, in capacitively transduced devices[30,31]. Dynamical mode coupling between gyroscopic modes is implemented by applying a red-detuned pump adjusted based on the system parameters and the built-in intrinsic coupling. This capacitive device is demonstrated to be a coupling-abundant multiple-mode system, which used to be very difficult to construct due to the difficulty of combining different types of mode coupling mechanisms in one system[40]. Strong dynamical coupling has significant practical applications to tuning the response of micro- and nanoelectromechanical devices, such as mode-localized sensors[41,42], Coriolis gyroscopes[43,44], wireless filter resonators[45], and many other devices[16,32].

## Results

**Capacitive symmetric electromechanical resonator**. A capacitive symmetric microelectromechanical ring resonator is used as the experimental testbed in this work[44]. The resonator is constructed of 45 equispaced nested rings. Adjacent rings are interconnected

with interleaved spokes. The resonator is supported by a central anchor. The diameter and the thickness of the resonator are 720 and 40 μm, respectively. The width of the rings and spokes is 3 μm. The resonator is surrounded by 16 capacitive electrodes, employed to actuate, sense, and tune the in-plane response. The capacitive gap $d_0$ is designed to be 1.5 μm wide. The device is fabricated using highly doped P-type (111) single-crystal silicon with a resistivity of 1–3 mΩ cm, and encapsulated using the "Episeal" process[46], enabling a stable <1 Pa vacuum environment. The capacitive electrodes introduce a nonlinear electrostatic field surrounding the mechanical resonator, which produces inhomogeneous force terms, stiffness modification terms, and nonlinear mode interactions among multiple modes. Combined with the intrinsic properties of the bare mechanical resonators, the capacitive features allows for abundant manipulations of the system dynamics, the most interesting aspect studied here is the parametric pump-induced dynamical operations.

The linear and nonlinear mode-coupling mechanisms are investigated in this system. Moreover, those couplings can be manipulated dynamically using a parametric pumping scheme. The experimental setup is shown in Fig. 1a; all the experiments are conducted at room temperature. The in-plane mechanical modes involved in this study are illustrated in Fig. 1b, c, involving two pairs of near-degenerate order-2 and order-3 modes, II-1, II-2, III-1, and III-2, with resonant angular frequencies $\omega_{\text{II-1}} = 2\pi \times 134, 209$ Hz, $\omega_{\text{II-2}} = 2\pi \times 134, 253$ Hz, $\omega_{\text{III-1}} = 2\pi \times 166, 498$ Hz, and $\omega_{\text{III-2}} = 2\pi \times 166, 949$ Hz. The damping rates of the modes of the same order are identical, with values of $\gamma_{\text{II}} \approx 2\pi \times 1.05$ Hz and $\gamma_{\text{III}} \approx 2\pi \times 2.51$ Hz. The electrodes are marked with numbers anticlockwise from 1 to 16 (Fig. 1a). The drive signals $\pm V_{\text{d}} \cos(\omega_{\text{d}} t)$ are applied on electrodes 3 and 7 to actuate order-2 modes. The pump signal $V_{\text{p}} \cos(\omega_{\text{p}} t)$ is applied on electrodes 1 and 5. A direct-current (DC) voltage $V_0$ of 30 V is applied on the resonator body. The mechanical motion is transduced by the detection electrodes along drive and pump axes. The current signal is introduced to a lock-in amplifier following further amplification stages. A tuning voltage $V_{\text{t1}}$ can be superposed on drive electrode 7. Another tuning voltage $V_{\text{t2}}$ can be applied on off-axis electrode 6.

**Hybrid state coupling and dynamical manipulation**. Order-2 modes II-1 and II-2 are a pair of widely used degenerate modes[47,48]. The antinodal axes (principal axes) of the order-2 normal modes have an angular interval of 45° (Fig. 2a). This system can be equivalent to a two-degree-of-freedom lumped parameter system in Cartesian coordinates (Fig. 2b). The angle in this equivalent system is the double of that in the real system depicted in Fig. 2a (see Supplementary Note 1). Axes $x$-$o$-$y$ are defined along the directions in which mechanical motions are probed, the drive and pump are applied along $x$ and $y$ directions, respectively. Coordinates $x_\omega$-$o$-$y_\omega$ are defined along the principal axes of normal modes II-1 ($x_\omega$ direction) and II-2 ($y_\omega$ direction).

If the tuning voltage $V_{\text{t1}}$ is applied along $x$ direction, the resonant frequencies of modes II-1 and II-2 will disperse (Fig. 2d). An avoided crossing is obviously illustrated when $V_{\text{t1}}$ is around 3.5 V, which indicates that the hybrid states observed from $x$-$o$-$y$ coordinates H-1 ($x$ direction) and H-2 ($y$ direction) can be regarded as two coupled resonators, as depicted in Fig. 2c. This is also revealed by the nondiagonal equations of motion in $x$-$o$-$y$ coordinates (see Supplementary Note 2). Those hybrid states are combinations of normal modes II-1 and II-2. The strength of the avoided crossing indicates the coupling rate of the two hybrid states. As shown in Fig. 2e, the $V_{\text{t1}}$ tuning process can be simulated (see Supplementary Note 2). This coupling originates from two factors. The first one is the frequency difference between the normal modes II-1 and II-2 ($\Delta\omega = \omega_{\text{II-2}} - \omega_{\text{II-1}} \neq 0$), and the

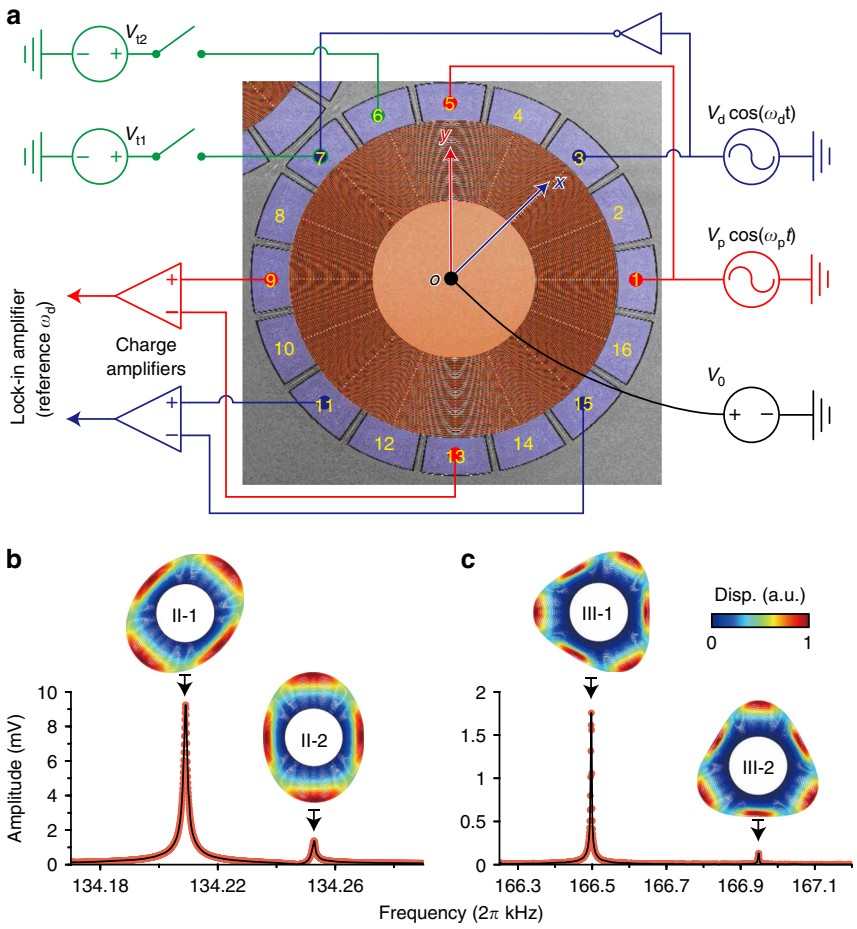

**Fig. 1 The experimental platform. a** Basic setup for experiments. The nested-ring resonator is actuated, pumped, tuned, and sensed by equally distributed capacitive electrodes numbered anticlockwise from 1 to 16. The drive signals are applied on electrodes 3 and 7 in the push–pull form. The pump signal is applied on electrodes 1 and 5. A DC voltage $V_0$ is applied on the resonator body. An in-axis tuning voltage $V_{t1}$ and an off-axis tuning voltage $V_{t2}$ are applied on electrodes 7 and 6, respectively. In the equivalent framework for order-2 degenerate modes, coordinates $x$ and $y$ are defined along drive and pump directions, respectively. The set of axes $x$-$o$-$y$ is of fourfold rotational symmetry. The response signal along $x$ ($y$) axis is detected by electrodes 11 (9) and 15 (13) differentially, amplified using charge amplifiers, and measured by a lock-in amplifier. **b, c** Mechanical modes involved in this study. **b** The amplitude–frequency responses and mode shapes of the order-2 in-plane modes, and **c** those of the order-3 in-plane modes. The displacements (disp.) are normalized. Source data are provided as a Source Data file

second one is the misalignment (denoted as $\theta$ in the equivalent Cartesian coordinates, as shown in Fig. 2b) of the principal axes with electrode axes, which are both due to manufacturing tolerances. The initial $\theta$ of the tested resonator is estimated to be about 18°. In the setup reference system, the real misalignment is $\theta/2$ (Fig. 2a). The mode shapes of the order-2 modes in the $V_{t1}$ tuning process can be obtained by calculating $\theta$ at different values of $V_{t1}$, which are selectively depicted in Fig. 2f. Both $\Delta\omega$ and $\theta$ can be changed by $V_{t1}$ or $V_{t2}$ (see Supplementary Note 2). A given set of $V_{t1}$ and $V_{t2}$ can determine a specific group of $\Delta\omega$ and $\theta$.

By regarding mode II-2 as a phonon cavity, a dynamical sideband coupling operation can be realized based on the above structural coupling by applying a red-detuned parametric pump. In this implementation, $V_{t1}$ and $V_{t2}$ are tuned to make $\theta \approx 36°$ and $\Delta\omega \approx 2\pi \times 39.7$ Hz. An alternating current signal $V_p \cos\omega_p t$ applied along $y$ axis will periodically change the stiffness of hybrid state H-2. The dynamics of the normal modes are given by the equations of motion

$$\ddot{x}_\omega + \gamma_{II}\dot{x}_\omega + (\omega_{II\text{-}1}^2 + \Delta_p\sin^2\theta)x_\omega + \Delta_p\cos\theta\sin\theta y_\omega = F_1\cos(\omega_d t), \quad (1)$$

$$\ddot{y}_\omega + \gamma_{II}\dot{y}_\omega + (\omega_{II\text{-}2}^2 + \Delta_p\cos^2\theta)y_\omega + \Delta_p\cos\theta\sin\theta x_\omega = F_2\cos(\omega_d t), \quad (2)$$

where $x_{II\text{-}j}$ and $F_j (j = 1, 2)$ are the displacement and drive force

amplitude (normalized by mass) of mode II-$j$, respectively. $\Delta_p$ is the pump produced by signal $V_p \cos(\omega_p t)$,

$$\Delta_p = \kappa\left[2V_0 V_p \cos(\omega_p t) - \frac{V_p^2}{2} - \frac{V_p^2}{2}\cos(2\omega_p t)\right], \quad (3)$$

where $\kappa = A_p\epsilon_0/(d_0^3 m_{II})$, $A_p$ is the area of pump electrodes, $\epsilon_0$ is the permittivity of vacuum, and $m_{II}$ denotes the effective mass of mode II-1 or II-2. There is a DC term, a first-order harmonic term, and a second-order harmonic term in the pump. The DC term in the pump will tune the hybrid coupling condition, thus affecting resonant frequencies (see Supplementary Fig. 6). The resonant frequencies of order-2 modes are functions of $V_p$ and are denoted as $\omega_{II\text{-}\square,V_p}$ ($\square = 1, 2$). When a 3-V pump is applied, $\omega_{II\text{-}1,3V}$ ($\omega_{II\text{-}2,3V}$) is slightly shifted from the initial $\omega_{II\text{-}1,0V}$ ($\omega_{II\text{-}2,0V}$) by 0.33 Hz (0.60 Hz), as shown in Fig. 3f.

The amplitude of the first-order term is $4V_0/V_p$ times that of the second-order term. The first-order term is dominant in this implementation, since $V_0$ is much larger than $V_p$. The second-order term itself and its interaction with the first-order term only slightly contribute to higher order ($\geq 2$) coupling. In the following interpretation parts, the DC and second-order harmonic terms

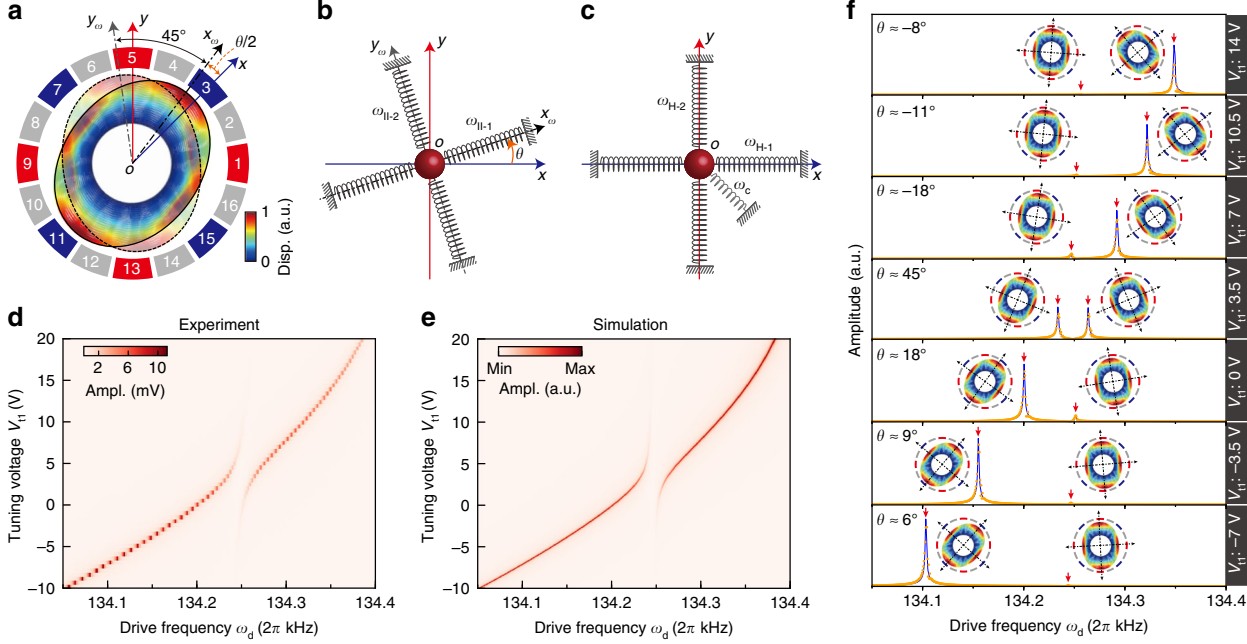

**Fig. 2** Hybrid state coupling caused by structural asymmetry. **a** Normal modes II-1 (solid) and II-2 (transparent) in the setup coordinates. $o\text{-}x_\omega$ and $o\text{-}y_\omega$ denote the principal axes of II-1 and II-2, respectively, both of which have the property of fourfold rotational symmetry. $o\text{-}x_\omega$ and $o\text{-}y_\omega$ have an intrinsic angular interval of 45°. **b** The order-2 normal modes are equivalent to an order-reduced two-degree-of-freedom lumped parameter system in Cartesian coordinates. $\theta$ is the angular offset between the equivalent probe coordinates $x\text{-}o\text{-}y$ and principal axes $x_\omega\text{-}o\text{-}y_\omega$. The as-fabricated $\theta$ of 18° is revealed, corresponding to the practical misalignment $\theta/2$ of 9° in **a**. **c** The mismatched system is equivalent to two hybrid states H-1 and H-2 along $x\text{-}o\text{-}y$ coupled by a crossing spring. **d** Amplitude (Ampl.) frequency responses detected along the x axis at $\omega_d$ with different values of $V_{t1}$; an avoided crossing is illustrated. **e** Simulation of electrostatic tuning in **d**. **f** Mode shape evolution when $V_{t1}$ is changed. Amplitude frequency responses include orange experimental points and blue theoretical fitting curves. The resonance peaks are marked with downward red arrows. In each mode-shape inset, black dot-dashed arrows indicate the principal axis of the normal mode. Source data are provided as a Source Data file

are not considered. $\Gamma_1 = 2\kappa V_0 V_p \sin^2\theta$ and $\Gamma_2 = 2\kappa V_0 V_p \cos^2\theta$ are defined as intra-modal coupling terms, and $\Lambda = 2\kappa V_0 V_p \cos\theta \sin\theta$ is defined as an inter-modal coupling term.

When simultaneously applying drive and red-detuned pump with frequencies $\omega_d \approx \omega_{II\text{-}1}$ and $\omega_p \approx \Delta\omega$, the first-order sideband dynamical coupling process is illustrated in Fig. 3a. An idler spectrum line near $\omega_{II\text{-}2}$ is generated due to an up-conversion ($\Lambda_+$) process, in which the anti-Stokes sideband 3 with frequency $\omega_{II\text{-}1} + \omega_p$ is produced in the dynamics of mode II-2 during the mixing of the inter-modal coupling pump with the displacement of mode II-1. Meanwhile, the mixing of mode II-2 displacement with the inter-modal coupling pump causes a down-conversion ($\Lambda_-$) process, which produces a Stokes sideband 2 with frequency $\omega_{II\text{-}2} - \omega_p$ in the dynamics of mode II-1. Sideband 2 has a phase delay relative to the external actuation tone. For the red-detuned pump, sideband 2 is in antiphase, which results in dynamic back-action cooling and avoided crossing. The off-resonance sideband 1 (4) is produced by the down (up)-conversion process of mode II-1 (II-2), which will not affect the dynamical coupling. In this implementation, sidebands are generated near the mechanical modes rather than the pumps in cavity optomechanics systems[49], because the eigen-frequencies of the mechanical modes are much higher than the frequency difference here.

If the pump frequency is changed to $\omega_p \approx \Delta\omega/2$, the second-order sideband coupling process is illustrated in Fig. 3b. Wave mixing processes of ($\Gamma_{1+} \times \Lambda_+$) and ($\Lambda_+ \times \Gamma_{2+}$) of mode II-1 will generate sidebands 7 and 8 in dynamics of mode II-2, respectively. Similarly, wave mixing processes of ($\Gamma_{2-} \times \Lambda_-$) and ($\Lambda_- \times \Gamma_{1-}$) of mode II-2 will generate sidebands 9 and 10 in dynamics of mode II-1. Off-resonance sidebands 5 and 6 act as the intermediary points for those processes. Other off-resonance

sidebands 11, 12, 13, 14 are also generated by first- and second-order mixing processes, which would not contribute to the dynamical coupling.

Sidebands 9 and 10 will produce back actions on mode II-1. However, the effects of sidebands 9 and 10 on mode II-1 are opposite to each other. Based on the second-order coupling strength model in Supplementary Eq. (58), back actions caused by sidebands 9 and 10 will completely cancel each other when $\theta = \arctan\sqrt{(3\omega_{II\text{-}1} + \omega_{II\text{-}2})/(\omega_{II\text{-}1} + 3\omega_{II\text{-}2})} \approx 45°$. In this condition, the second-order dynamical coupling is invisible, as shown in Supplementary Fig. 3a.

Here, we monitor the frequency response along the x axis at $\omega_d$, and the results with different pump frequencies are shown in Fig. 3c. The pump strength is maintained at $V_p = 3$ V. The vertical dotted lines indicate the pump-on resonant frequencies of the modes II-1 and II-2. First level normal-mode splitting can be observed when $\omega_p \approx \Delta\omega$, which indicates strong first-order dynamical mode coupling. The avoided crossings induced by the very strong first-order dynamical coupling shift the lower branches of the resonance peaks, which makes the second-order avoided crossings take place at a pump frequency higher than $\Delta\omega/2$, as shown in Fig. 3c, d. However, it should be noted that the second-order dynamical coupling still takes place at the pump frequency of exactly $\Delta\omega/2$. The observed second-order avoided crossings higher than $\Delta\omega/2$ are simultaneously affected by both the first- and the second-order coupling. Some slices of response curves near and between the first- and second-order coupling are also provided in Fig. 3e to better illustrate the avoided crossing evolution process. The Stückelberg interferometry pattern is also observed at the bottom of the Fig. 3c where the pump frequency is low[50]. In this system, the intra-modal coupling terms are

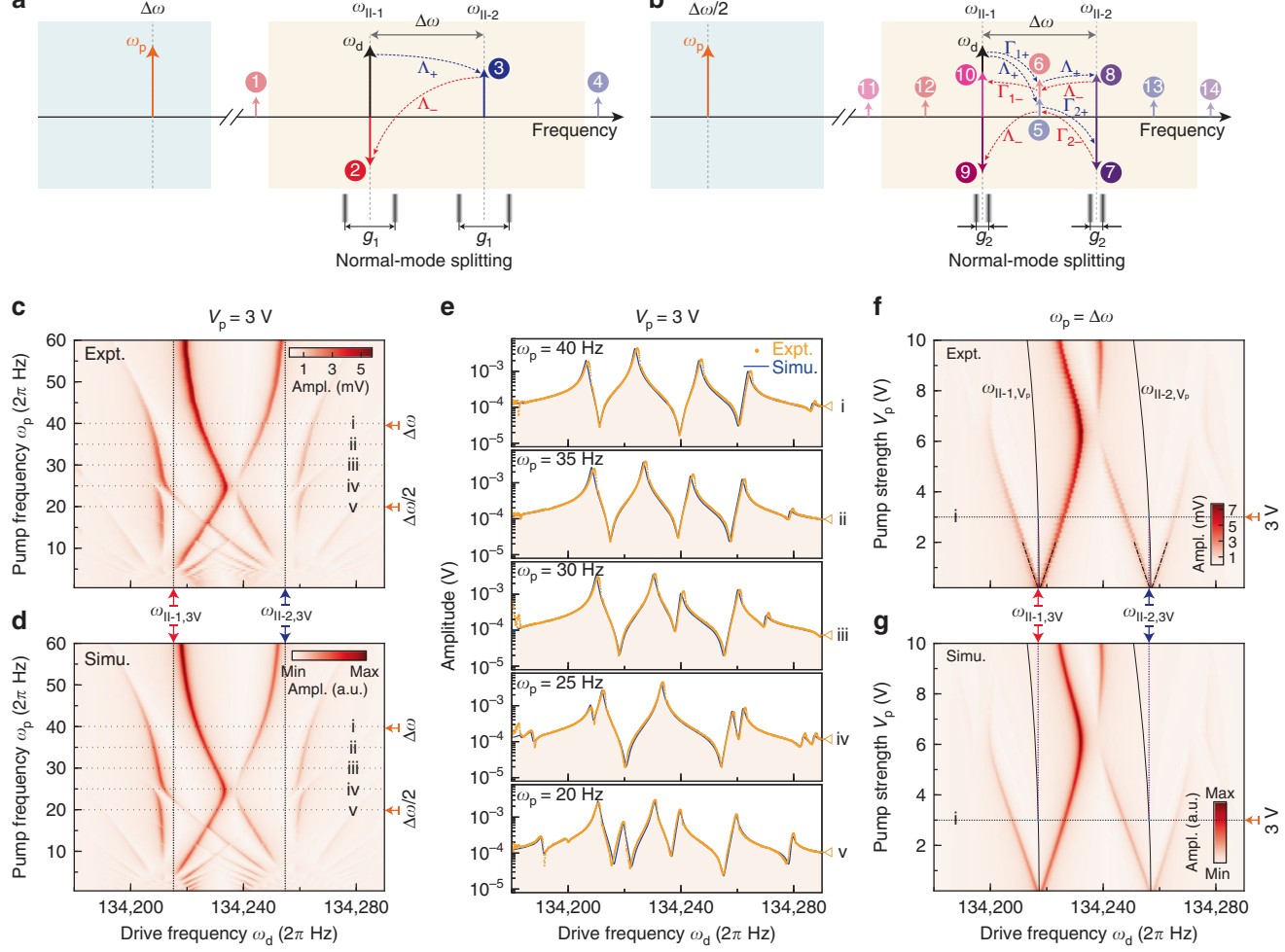

**Fig. 3** Dynamical coupling between modes II-1 and II-2. **a** Schematic spectrum of the first-order sideband coupling. The red-detuned pump is applied at $\Delta\omega$ (orange tone), and external force is actuating mode II-1 (black tone). The light blue (orange) region indicates the sweeping range of the pump (drive) frequency in **c**. $\Lambda_+$ and $\Lambda_-$ indicate inter-modal up- and down-conversion processes, respectively. Sideband 2 is in antiphase with respect to the drive tone. Normal-mode splitting takes place if first-order pump is applied, and the frequency split indicates the first-order coupling rate $g_1$. **b** Schematic spectrum of the second-order sideband coupling. The red-detuned pump is applied at $\Delta\omega/2$ (orange tone), and external force is actuating at mode II-1 (black tone). $\Gamma_{i+}$ and $\Gamma_{i-}$ ($i = 1, 2$) indicate intra-modal up- and down-conversion processes, respectively. Sideband 9 (10) is in antiphase (phase) with respect to the drive tone. Sideband 7 is in antiphase with respect to sideband 8. Normal-mode splitting takes place if the second-order pump is applied, and the frequency split indicates the second-order coupling rate $g_2$. It should be noted that the phase relations between different frequency tones are not depicted in **a** and **b**. **c** Drive frequency $\omega_d$ and pump frequency $\omega_p$ response along $x$ axis detected at $\omega_d$ when $V_p = 3$ V. The vertical dotted lines indicate the mode frequencies $\omega_{\text{II-}\square,3V}$ ($\square = 1, 2$) affected by the DC term in the 3-V pump. Expt. indicates experimental data. **d** Simulation of the experimental results in **c** by solving Eqs. (1), (2), parameters of $\theta = 36°$ and $\Delta\omega = 2\pi \times 39.7$ Hz are used in the simulation. Simu. indicates simulation results. **e** Some slices of frequency response curves in **c** and **d**, which are denoted by numbers i–v. The orange points are experimental data and blue curves are simulation results. **f** The frequency response along $x$ axis as a function of pump amplitude $V_p$ when $\omega_p = \Delta\omega$. The DC term in the pump will slightly shift resonant frequencies (black solid curves) $\omega_{\text{II-}\square,V_p}$ to lower values. **g** Simulation of the experimental results in **f**. Source data are provided as a Source Data file

responsible for the Stückelberg interferometry phenomenon, and the inter-modal coupling term leads to the first-order dynamical coupling. Both intra-and inter-modal coupling terms are necessary for higher-order dynamical couplings. The experimental data can be reproduced by solving Eqs. (1), (2), as shown in Fig. 3d (see Supplementary Note 3).

The dynamical coupling strength is controllable. Fig. 3f shows the pump strength $V_p$ dependence of the first-order coupling rate $g_1$, which is described by the frequency split. The first-order coupling rate is given by $g_1 = \sqrt{\kappa^2 V_0^2 V_p^2 \sin^2(2\theta)/(4\omega_{\text{II-1}}\omega_{\text{II-2}}) - \gamma_{\text{II}}^2}$. The theoretical values of mode splitting are depicted by dot-dashed lines in Fig. 3f. In this study, first-order coupling rate of more than $2\pi \times 30$ Hz can be obtained, which exceeds the damping rate

$(2\pi \times 1.05$ Hz) by more than 28 times. Besides, when $V_p$ reaches values around 6 V, an intermediate avoided crossing is observed, which originates from the interaction of the two first-order idler sideband. If $V_p$ exceeds this value, the frequency split will decrease. The maximum coupling rate of this study is restricted by the limited value of $\Delta\omega$. It is noteworthy that the dynamical coupling strengths are both $V_p$ and $\theta$ dependent. Adjusting $\theta$ provides a new degree of freedom to control dynamical coupling.

The pump strength dependence of the first-order coupling strength can be reproduced by solving Eqs. (1), (2) (Fig. 3g). There is an overall trend of dispersing to lower frequency for the peaks in Fig. 3f, which is caused by the DC term in the pump $\Delta_p$. The experiment and simulation results for another

implementation of $\theta \approx 45°$ and $\Delta\omega \approx 2\pi \times 40.7$ Hz are also provided in Supplementary Fig. 3.

**Electrostatic nonlinear coupling and dynamical manipulation.** In this setup, the ring resonator is dominated by stiffness-softening electrostatic nonlinearity because of the narrow capacitive gap[51], which is confirmed by the $V_0$-dependence of the nonlinear response (see Supplementary Note 4). Doping-induced material nonlinearity[52] and tension-induced mechanical nonlinearity[51] may also reside, but they are relatively weak compared to the electrostatic one when $V_0$ is set to be 30 V.

It was previously demonstrated that the mechanical modes in clamped–clamped beam resonators can be parametrically coupled to each other due to tension-induced mechanical nonlinearity, because the displacement of one mode produces a beam extension thus modifying the other mode's resonant frequency[18,22,23]. Here, we demonstrate that the origin of electrostatic nonlinearity can also produce intermodal parametric coupling. The electrostatic nonlinear coupling between order-2 and order-3 modes is experimentally and theoretically described.

Modes II-1 (II-2) and III-1 are simultaneously actuated by drive and pump electrodes, respectively. Modes II are actuated in linear region with a drive signal amplitude of 2 mV. Mode III-1 is actuated in the stiffness-softening Duffing nonlinear condition with a drive signal amplitude of 10 mV. The frequency responses of modes II are recorded, and a dispersive parametric coupling is observed. If the drive frequency of mode III-1 is changed from low to high, the dispersion of modes II is shown in Fig. 4d. If the drive frequency of mode III-1 is changed from high to low, the dispersion of modes II is shown in Fig. 4e, and the shifts of modes II-1 and II-2 when mode III-1 is at resonance are 5.9 Hz and 6.2 Hz, respectively. The nonlinear bifurcation of mode III-1 is revealed by the frequency dispersion of modes II. When mode III-1 is actuated, the resonant frequencies of modes II will disperse to lower values. This phenomenon is opposite to that of the mechanical nonlinear parametric coupling[18,22,23], which should make modes II disperse to higher resonance frequencies.

The observed parametric mode coupling can be explained by the model that two modes sharing one polarized capacitor. It was previously shown that a mechanical resonator can be coupled to a microwave cavity using a capacitor[10,53,54], and the stiffness of the mechanical resonator is modified by varying the bias voltage on the capacitor. In this model, the resonant frequency of mode II is modified by the capacitance gap variation induced by displacement of mode III. When two modes are actuated simultaneously, their displacements are superposed (Fig. 4a–c). Displacement of heavily actuated mode III will cause a electrostatic stiffness variation for mode II, thus modifying its resonant frequency. The capacitive parametric coupling can be described by following equations of motion (see Supplementary Note 4):

$$\ddot{x}_{II} + \gamma_{II}\dot{x}_{II} + \omega_{II}^2 x_{II} + \alpha_{II}x_{III} + \beta_{II}(x_{II} + x_{III})^2 \\ + \nu_{II}(x_{II} + x_{III})^3 = F_{II}\sin(\omega_{d\text{-}II}t), \tag{4}$$

$$\ddot{x}_{III} + \gamma_{III}\dot{x}_{III} + \omega_{III}^2 x_{III} + \alpha_{III}x_{II} + \beta_{III}(x_{II} + x_{III})^2 \\ + \nu_{III}(x_{II} + x_{III})^3 = F_{III}\sin(\omega_{d\text{-}III}t), \tag{5}$$

where $x_{II}$ and $\omega_{II}$ denote the displacement and resonant angular frequency of mode II-1 or II-2, and $x_{III}$ and $\omega_{III}$ denote the displacement and resonant angular frequency of one order-3 mode (mode III-1 in this case). $F_\square$ and $\omega_{d\text{-}\square}$ ($\square =$ II, III) are the amplitudes and frequencies of the external forces acting on the corresponding modes. Parameters $\alpha_\square$, $\beta_\square$, $\nu_\square$ are provided by Supplementary Eqs. (70–72). The key factor for the observed mode-II frequency shift is the $3\nu_{II}x_{III}^2 x_{II}$ term in expanded Eq. (4).

The displacement square of mode-III scaled by the third-order nonlinearity coefficient $\nu_{II}$ will directly influence the effective stiffness of mode-II. Eqs. (4), (5) are solved using multiple-scale analysis, and the results are provided in Supplementary Eqs. (75, 76).

The dispersive frequency shifts of modes II caused by actuation of mode III-1 can be simulated based on Eqs. (4), (5) (see Supplementary Fig. 9). The frequency shift of mode II $\hat{\sigma}_{II}$ caused by displacement of mode III and that of mode III $\hat{\sigma}_{III}$ caused by displacement of mode II are given by

$$\hat{\sigma}_{II} \approx \frac{3\nu_{II}d_0^2\gamma_{II}}{8\omega_{II}}\left(\frac{f_{II}^2}{\omega_{II}^2} + \frac{2f_{III}^2\gamma_{II}^2}{\omega_{III}^2\gamma_{III}^2}\right), \tag{6}$$

$$\hat{\sigma}_{III} \approx \frac{3\nu_{III}d_0^2\gamma_{II}}{8\omega_{III}}\left(\frac{f_{III}^2\gamma_{II}^2}{\omega_{III}^2\gamma_{III}^2} + \frac{2f_{II}^2}{\omega_{II}^2}\right), \tag{7}$$

where $f_\square = F_\square/(d_0\gamma_{II}^2)$. The frequency shift direction of mode-II (the sign of $\hat{\sigma}_{II}$) is determined by the sign of the third-order nonlinearity coefficient $\nu_{II}$, which is negative for this resonator dominated by electrostatic nonlinearity. Thus, frequency of mode-II will shift downward.

As long as the associated modes can simultaneously modulate the response of a common capacitive transducer, those modes are coupled. Thus, this electrostatic nonlinear mode coupling is very common in capacitive micro- or nanomechanical resonators. The coupling strength is significantly impacted by the characteristics of the shared capacitor.

It has been illustrated that the structural asymmetry will transform normal modes II-1 and II-2 into mechanically coupled hybrid states H-1 and H-2. The hybrid states are electrostatically coupled to order-3 modes, an anti-Stokes pump can transform this electrostatic coupling into tunable strong dynamical coupling. In this case, order-3 modes are regarded as phonon cavities. The drive signal ($V_d = 2$ mV) sweeps from below mode II-1 to above mode II-2, and the pump signal ($V_p = 3$ V) with frequency from below $\omega_{III\text{-}1} - \omega_{II\text{-}2}$ to above $\omega_{III\text{-}1} - \omega_{II\text{-}1}$ are simultaneously applied. The frequency responses of modes II-1 and II-2 are measured, as shown in Fig. 4f, in which a skewed "#" configuration with four avoided crossings is illustrated. When the pump frequency is kept $\omega_{III\text{-}1} - \omega_{II\text{-}1}$, the dynamical coupling strength of modes II-1 and III-1 depicted by the frequency split of mode II-1 will increase if pump voltage is increased (Fig. 4h). The pump voltage dependence of coupling strength of modes II-2 and III-1 is also obtained (Fig. 4j). Moreover, avoided crossings can be observed between the higher splitting branch of II-1 and mode II-2 in Fig. 4h, and between the lower splitting branch of II-2 and mode II-1 in Fig. 4j, which are produced by the structural hybrid coupling of H-1 and H-2.

The skewed "#" coupling configuration are simulated by separately modeling the sequential dynamical coupling of III-1 to II-1 and II-2 and that of III-2 to II-1 and II-2, which can be described similarly by the following equations of motion:

$$\ddot{x}_{II\text{-}1} + \gamma_{II}\dot{x}_{II\text{-}1} + \omega_{II\text{-}1}^2 x_{II\text{-}1} + \alpha_1 x_{II\text{-}1} + \beta_1 x_{II\text{-}2} + \lambda_1 x_{III} \\ + \Lambda_1\cos(\omega_p t)(x_{II\text{-}1}\sin\theta + x_{II\text{-}2}\cos\theta + x_{III}) \\ = g_1\cos(\omega_d t), \tag{8}$$

$$\ddot{x}_{II\text{-}2} + \gamma_{II}\dot{x}_{II\text{-}2} + \omega_{II\text{-}2}^2 x_{II\text{-}2} + \alpha_2 x_{II\text{-}1} + \beta_2 x_{II\text{-}2} + \lambda_2 x_{III} \\ + \Lambda_2\cos(\omega_p t)(x_{II\text{-}1}\sin\theta + x_{II\text{-}2}\cos\theta + x_{III}) \\ = g_2\cos(\omega_d t), \tag{9}$$

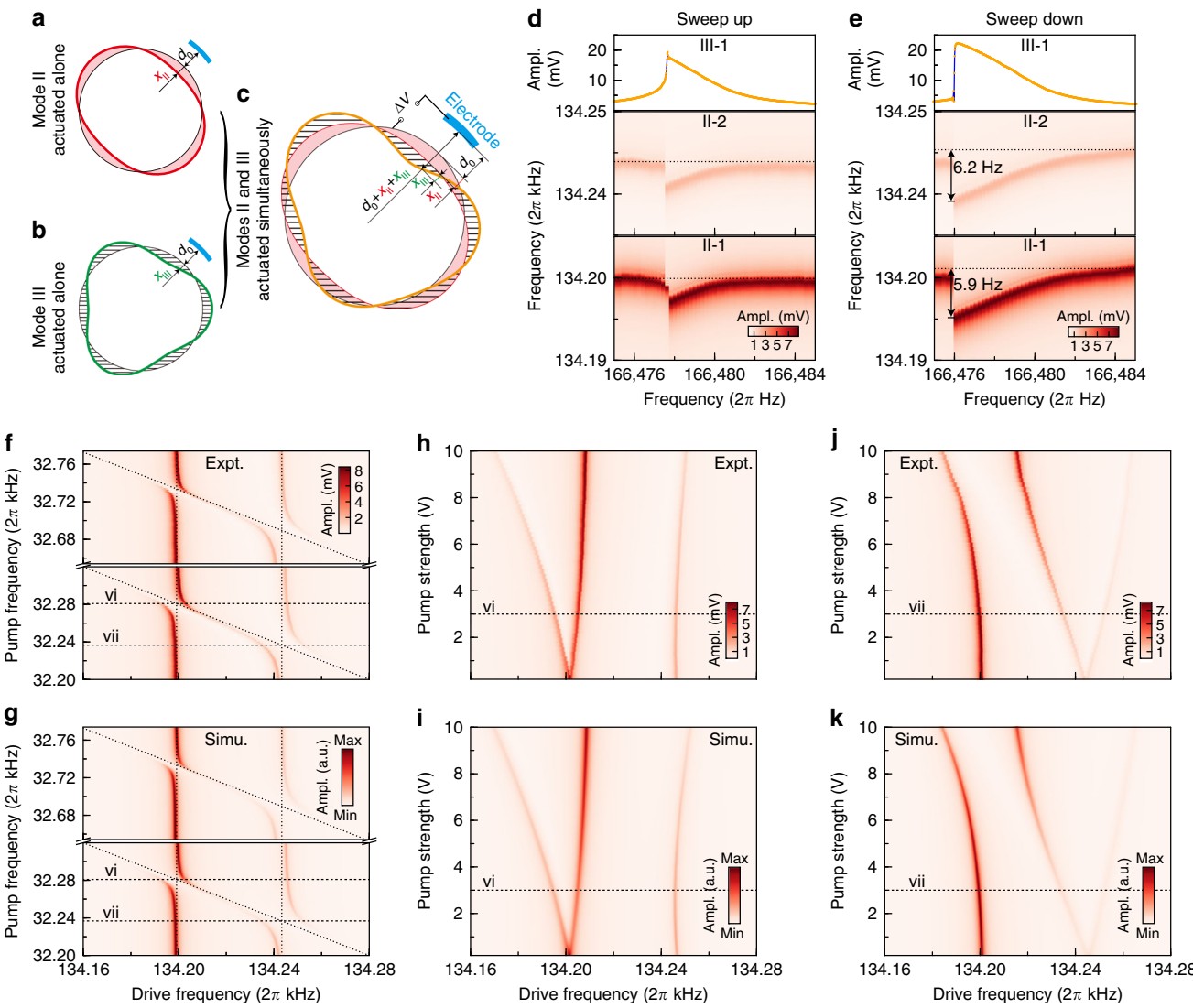

**Fig. 4** Electrostatic nonlinear parametric coupling and demonstration of coupling-abundant multiple-mode system. **a**, **b** Schematic transient pattern of the independently actuated mode II (**a**) and mode III (**b**). **c** Schematic transient pattern of the simultaneously actuated modes. Their superposed displacements will both affect the capacitive gap. Modal interaction occurs when a nonlinear electrostatic potential is applied. **d**, **e** The dispersive frequency shifts of modes II induced by the actuation of mode III-1 if the drive frequency of mode III-1 is changed from low to high (**d**) and high to low (**e**). The dotted lines indicate the resonant frequencies of the modes II without the additional actuation of mode III-1. The bifurcation-induced jump phenomenon in nonlinear frequency response of mode III-1 (upside) can be detected by the frequency shift of modes II. **f** Frequency responses of modes II when pump frequency are changed from below $\omega_{III-1} - \omega_{II-2}$ to above $\omega_{II-2} - \omega_{II-1}$ depict a skewed "#" configuration. The pump voltage is 3 V. **g** Simulation of the experimental results in **f**. **h** The strength of the dynamical coupling between modes II-1 and III-1 as a function of pump amplitude $V_p$, pump frequency is set to be $\omega_{III-1} - \omega_{II-1}$. **i** Simulation of the experimental results in **h**. **j** The strength of the dynamical coupling between modes II-2 and III-1 as a function of pump amplitude $V_p$, pump frequency is set to be $\omega_{III-1} - \omega_{II-2}$. **k** Simulation of the experimental results in **j**. Source data are provided as a Source Data file

$$
\ddot{x}_{III} + \gamma_{III}\dot{x}_{III} + \omega_{III}^2 x_{III} + \alpha_3 x_{II-1} + \beta_3 x_{II-2} + \lambda_3 x_{III} \\
+ \Lambda_3 \cos(\omega_p t)(x_{II-1}\sin\theta + x_{II-2}\cos\theta + x_{III}) = 0. \quad (10)
$$

The subscript III refers to mode III-1 or III-2. The detailed derivation of those equations of motion and the parameters $\alpha_j$, $\beta_j$, $\lambda_j$, $\Lambda_j$, and $g_j$ ($j = 1, 2, 3$) are provided in Supplementary Note 5. By solving Eqs. (8)–(10), we can simulate the observed avoided crossings, as shown in Fig. 4g. The $V_p$ dependence of coupling strengths depicted in Fig. 4h, j can also be simulated by setting $\omega_p = \omega_{III-1} - \omega_{II-1}$ and $\omega_{III-1} - \omega_{II-2}$, as shown in Fig. 4i, k, respectively.

By showing the abundant coupling phenomena between order-2 and order-3 modes, we demonstrate that this

electrostatic mechanical system has the potential to couple even more modes. If the dynamical couplings are implemented simultaneously, a classical analog of multiple-level system could be constructed, which would enable abundant varieties of interesting manipulations[40].

## Discussion

In this study, we report significant progress in modeling and manipulating the structural asymmetry and misalignment induced mode coupling between a pair of degenerate wine-glass modes in ring resonators. We also discover an electrostatic field-imposed nonlinear parametric coupling effect among different modes in a single resonator. These modal coupling effects in

capacitively transduced mechanical systems can be tuned very significantly through the parametric pumping scheme. Electrostatic-pump-based manipulations show consequent advantages with respect to tension-mediated manipulations. Depending on the topology of the resonators, the effect of tension-induced manipulations can vary. For instance, its effect can be significant for clamped–clamped or thin film resonators, but negligible for centrally anchored or bulk-fabrication-process resonators, such as the ring resonator in this work, if the same tension is applied. In comparison, the electrostatic-mediated pump-based manipulation removes such topology restrictions for capacitively transduced resonators, hence making it applicable for a wider range of resonator designs.

Recently, significant breakthroughs in enhancing quality factor ($Q$) of mechanical resonators have been made[55–58]. Though some of those resonators are dielectric, the proposed dissipation mitigating techniques could inspire conductive high-$Q$ mechanical resonators. An interesting direction for further research is to combine the electrostatic-mediated parametric coupling and dynamical manipulations illustrated here in such conductive high-$Q$ mechanical resonators, which may enable purely phonon-based quantum sideband manipulations at the macroscopic scale[10,11]. Some interesting dynamical tunable coupling experiments have been implemented in capacitive mechanical systems[30,31]. However, these effects were attributed to tension-induced parametric interaction. Here, we demonstrate that electrostatic field-induced parametric coupling should also reside in these systems, and may at least partly contribute to the observed results.

Electrostatic mechanical resonators are widely used for sensing applications, and their susceptibility to mode coupling has often been not fully assessed in published studies. This paper shows that mode coupling should be taken into account when designing sensors, as the associated interactions may greatly alter the properties of the mechanical devices. More interestingly, electrostatic dynamical sideband coupling may be very useful in terms of enhancing sensor performance, such as manipulating energy transfer between dynamically coupled modes to improve or decrease $Q$ factor of a specific mode[59], greatly speeding up mode switching for Coriolis gyroscopes by replacing the stop-decay-actuate process with coherent mode manipulation[60], thus calibrating the bias error without affecting the bandwidth, dynamically tuning the frequency split of gyroscopes, and providing tunable coupling for mode-localized sensors.

## Methods

**Experiment and simulation setup.** The die embedding the vacuum-sealed MEMS resonators is electrically packaged in a ceramic leadless chip carrier. The device is operated in an ambient room temperature environment. The bias voltages are generated by a low noise voltage source (Keysight B2961A). The drive and pump signals are provided by a two-channel lock-in amplifier (Zurich Instruments HF2LI). The response motion of the resonator is detected by a capacitance–voltage converting scheme that is based on charge amplifier, and measured by the lock-in amplifier. The simulation codes are based on Python 3.7 with NumPy and Matplotlib packages.

## Data availability

The authors declare that all data supporting the findings of this study are included in the paper and its supplementary information files, and are available on request from the corresponding authors. The source data underlying Figs. 1b, 2d, f, 3c, e, f, and 4d, e, f, h, j and Supplementary Figs. 2c, 3a, c, e, 4c–i, 5, 6, 7, and 8 are provided as a Source Data file, which is also available in figshare (https://doi.org/10.6084/m9.figshare.8397908).

## Code availability

The authors declare that all codes supporting the findings of this study are included in the paper and its supplementary information files, and are available on request from the corresponding authors. The simulation codes underlying Figs. 2e, 3d, g, and 4g, i, k and Supplementary Figs. 2d, e, 3b, d, f, 4c–i, 6, and 9 are available in figshare (https://doi.org/10.6084/m9.figshare.8397908).

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

## Acknowledgements
We thank Sijun Du, Milind Pandit, Malar Chellasivalingam, and Atif Aziz for helpful discussions and assisting with training on laboratory equipment. This work is partly supported by the National Key R&D Program of China (NKPs) (2018YFB2002304) and the National Natural Science Foundation of China (NSFC) (51905539, 51575521 and 51705527). Experimental devices are designed and fabricated in the nano@Stanford labs, which are supported by the NSF as part of the National Nanotechnology Coordinated Infrastructure under award ECCS-1542152, with support from the Defense Advanced Research Projects Agency Precise Robust Inertial Guidance for Munitions (PRIGM) Program, managed by Dr. Robert Lutwak and Dr. Ron Polcawich, and the NSF under grant number CMMI-1662464. This work is primarily supported by the UK Natural Environment Research Council under grant number NE/N012097/1.

## Author contributions
X.Z. conceived the idea and designed the research under the guidance of A.A.S. and assistance of C.Z. The measurements and data analyses were performed by X.Z. and C.Z. with the assistance of J.S. and G.S. The test circuitry was developed by D.X. and X.Z. The theoretical works were done by X.Z., supervised by A.A.S., D.X. and X.W. The device was designed and fabricated by D.D.G., Y.C., I.F. and T.W.K. The manuscript was written by X.Z. and A.A.S. All authors contributed to manuscript preparation. The project was planned by A.A.S.

## Competing interests
The authors declare no competing interests.

## Additional information

**Peer Review Information** *Nature Communications* thanks G. P. Guo, Zenghui Wang and other anonymous reviewer(s) for their contribution to the peer review of this work. Peer reviewer reports are available.

