## [Peer Review File · Nature Communications]

Reviewers' comments:

Reviewer #1 (Remarks to the Author):

The manuscript describes the tuning of mode coupling in a MEMS resonator through electrostatic control. A number of different coupling behaviors are observed and carefully analyzed, and the analytical results beautifully match the experimental data. The extent of the work and the quality of results are of great potential to make a good publication; however, a number of issues should be addressed before this manuscript is fully suitable for publication in Nature Communications.

The authors mentioned a number of tunable mode coupling work. It would be interesting to compare the electrostatic-tuned mechanism reported in this manuscript with, say, tension-induced ones, such as those in references 27, 29, 30, and Nano Lett. 15, 6727 (2015), and articulate the unique advantages of electrostatic tuning demonstrated in this work.

The authors showed that the modes II-1 and II-2 can be measured and tuned using different electrodes. Here, it was determined that the two intrinsic II modes (without any tuning), whose frequencies split due to asymmetries due to fabrication process, have mode shapes that are 18 degrees off from the x and y axis in the setup coordinate. Why didn't the authors choose to adjust their measurement arrangement (by connecting to different electrodes) so that the electrodes are best aligned with the intrinsic mode shapes? Since the authors have electrode every 22.5 degrees, it seems natural to rotate all the connections by one electrode so that misalignment between drive/probe and the intrinsic mode shape is off by only 4.5 degrees.

It would be helpful to label some of the angles in Fig. 1 so the readers can more easily relate to the description in the text.

Since in the actual measurements the authors choose to use the measurement arrangement as it is, it may be more intuitive to show the mode shapes with their actual orientation in Fig. 1b (with theta also shown), which will help the readers better understand the part of the misalignment between the mode shape and the electrodes.

The authors used V_{t1} and V_{t2} to tune the frequencies of the II-1 and II-2 modes. Since these two mode shapes are 45 degrees from each other in real space, it would be intuitive to use two electrodes that are also 45 degrees apart to tune these two modes (as the respective nodes and antinodes in the two modes are also 45 degrees apart). Why do the authors choose to use two electrodes that are only 22.5 degrees apart? It seems a bit counterintuitive—suppose one electrode tunes one mode most effectively, then the other electrode would be sitting almost in between the two mode shapes, and would likely be tuning both II modes together?

I do notice that tuning using V_{t1} (with measurement both along x and y axes are shown in Fig. 2b and S1c), in which clearly one mode is being effectively tuned from 134.0 to 134.4 kHz, while the other mode is mostly intact. I wonder what happens when the similar experiment is repeated for V_{t2} —do both modes move together? Or away from each other? Since it would be unlikely that V_{t2} can

effectively tune just one mode and leave the other one intact due to the reason in the previous paragraph.

Meanwhile, it would be intuitive to illustrate this process (tuning using V_{t1}) using similar presentation as in Fig. 1b to supplement the data in Fig. 2b, by showing the frequency response curve AND mode shapes as V_{t1} increases, so the readers can better understand how II-1 and II-2 gradually evolve into H-1 and H-2; how does the mode shape of H-1 and H-2 look like; what does the mode shape look like during anti-crossing, etc. Therefore I invite the authors to seriously consider moving Fig. 2a,b,c into a new figure, and adding a few line plots of frequency response with simulated mode shape illustrations as in Fig. 1b, with corresponding V_{t1} values outlined on the color plot (current Fig. 2b), so help readers comprehend this process better.

After equation 2, the authors mentioned that “The amplitude of the first-order term is $2V_0/V_p$ times that of the second-order term”. Please double check the math and see if it would be 4 instead of 2.

For the application of the pump signal, there is a lot information shown in Fig. 2d-i, and in the current presentation it is rather dense and a bit challenging to follow for most readers. I suggest the authors considering making it easier for the average reader by doing the following:

- Make Fig. 2d-i a separate figure of its own. Actually with Fig. 2a-c moved out this should have been achieved already.
- Align Fig. 2d with 2f,g vertically on the frequency axis. The authors might consider reduce the span of the low frequency part a bit (to the left of the axis break), and make the high frequency part (to the right of the axis break) exactly aligned with the color plots.
- Use thin vertical dashed lines through 2d,f,g to indicate the drive, idler, and sidebands (1-4).
- Add a horizontal dashed line in 2f to indicate $\omega_p = \Delta\omega$, so that the readers can see which horizontal slice in fig. 2f corresponds to Fig. 2d.
- Similarly align Fig. 2e with Fig. 2f and 2g (maybe from underneath, or in between the two, depending on which arrangement is best for presentation). Draw similar vertical dashed lines for bands 5-14 in Fig. 2e.
- Add a horizontal dashed line in 2f or 2g to indicate $\omega_p = 1/2 \Delta\omega$, so that the readers can see which horizontal slice in fig. 2f or 2g corresponds to Fig. 2e.
- Use horizontal dashed lines in Fig. 2f and/or 2g to indicate where the pump strength sweep in Fig. 2h,i is conducted.

Taking these measures can help better illustrate the experiment process, and allow the readers to quickly understand the experiment data.

When discussing the nonlinearity in the resonator, the authors mentioned that “It is also known that the doping process impacts material nonlinearity [51], which is often dominant in bulk-mode resonators. However, flexural resonators such as the ring resonator in this study [51] are limited by nonlinear effects due to the imposed electrostatic field.” Why material nonlinearity is not important in this resonator under study? It is unclear from the writing.

In Fig. 2b,c, why is the legend arranged with 2mV on top and 10mV at bottom? That arrangement seems counterintuitive and completely opposite from that of the actual data (making those print on black-and-white really hard to follow)

In Fig. 2d,e color plots, it would be helpful to label II-1 and II-2 modes.

For the experiment in Fig. 3f, it would be helpful to use vertical dashed lines to indicate II-1, II-2, and a sloped dashed line to show (III-1)-(ω_p), so the readers to relate to the features on the figure. If possible, it would also be helpful to show the detected signal for mode III-1 at its frequency when sweeping ω_p as in Fig. 3f, to show the power is being transferred from the II modes to the III-1 mode.

Similar to the II case, it would be helpful to use horizontal dashed line in Fig. 3f to indicate where the experiments in fig. 3h,j are conducted. Also what's the reason for the apparent splitting observed in 3h and 3j? It was clear from the current writing.

I am not sure if the nonlinearity shown in Fig. 3b-e is necessary for observing the mode coupling in Fig. 3f-k. From the color scale in Fig. 2f it seems the authors are actuating II-1 only using 2mV (the smallest one in Fig. 2c), and still clearly observed the mode coupling. Then one would conclude the nonlinearity shown in Fig. 2b,c are not necessary for observing the mode coupling in Fig. 3f-k. In that case, maybe the authors can re-arrange the order and tell the story of Fig. 3f-k first? As that part is more in the same line with the story in the previous figure. And Fig. 3b-e may be moved to the end as a separate side story, or into the SI, as it does not affect the main story line of the main text.

The authors claimed that "In this case, hybrid states H-1, H-2, and mode III-1 construct a coupled-three-mode configuration". However, in the experiment shown in Fig. 3f, by sweeping ω_p mode III-1 is sequentially coupled to II-1 and II-2, but the responses of II-1 and II-2 never cross each other. So it appears what the authors have demonstrated are two separate two-mode couplings II-1 to III-1, and II-2 to III-1, in the same measurement, which can hardly be called a "coupled-three-mode"—because if one extends the range of ω_p a bit more III-2 mode would also be involved—would one call it "coupled-four-mode" then (and one would imagine the data in 3f would look like a skewed "#")? There is no limit to how many mode coupling one could observe in such experiments, but each crossing (or avoided crossing) represents the coupling between two individual modes—unless one tune the frequency and make 3 distinct modes cross at the same point. In fact, the authors may consider including data in Fig. 3f with ω_p go from below III-1 to above III-2 to show additional couplings, which would be more illustrative.

A number of typos should also be fixed before resubmission, such as "anhiliation".

It would be helpful to show the detailed structure (such as SEM images) of the resonator in SI.

Overall this is a solid piece of work, and a number of interesting observations are being made with good match between numerical model and experiment data. While the current presentation is a bit too dense and difficult to follow by average readers, after appropriate and sufficient improvement I will re-consider recommendation of its publication.

Reviewer #2 (Remarks to the Author):

The manuscript has investigated sideband coupling in a silicon-based gyroscopic ring resonator. Parametric coupling is demonstrated in both two-mode and three-mode configurations. Overall, the paper is well-written and well-organized. However, I cannot recommend its publication in Nature Communications at present based on the following comments:

1. I'm wondering what is the difference between this paper and other works on parametric coupling in micro-beam or micro-cantilever based mechanical resonators. For example, H. Okamoto et al. have studied dynamic coupling and observed high-order process under similar periodic pumping set-up (see Nature Physics 9, 480 (2013)). Can the authors further illustrate the novelty of this manuscript?
2. The authors claim that there is an initial misalignment θ , induced by manufacturing process. Also, under some particular value, for example $\theta=45$ degree, the coupling pattern is changed (Figure S2). Is it possible to tune the pattern in Figure 2f to that in Figure S2c to cancel second-order coupling, since θ can be tuned electrically by changing V_{t1} and V_{t2} ?
3. About the coupling mechanism in Figure 3a, what is the shared capacitance? Is it a real capacitance (or sum of some real capacitances) or an imaginary effective capacitance? What is the physical consideration behind this model?
4. In Figure 2h, the frequency response seems to be asymmetric when pump strength is larger than 6V, after two first-order response crossing each other. Similarly, the frequency response of mode II-1 becomes asymmetric from the very beginning at around pump strength=1V in Figure 3h. Any explanations about this?
5. Several misspelling: for example anihilation in Page 4, conurbation in captions of Figure 3.

Overall, I cannot recommend publication of this manuscript before all the comments being addressed.

Reviewer #3 (Remarks to the Author):

This manuscript investigates modal coupling in microelectromechanical gyroscopic ring resonators. The authors demonstrate precise control the mode coupling by observation of avoided mode crossings under dynamic capacitive tuning. In addition, they studied dynamical coupling effects using a nonlinear parametric coupling. Detailed analyses, simulations, and experiments were presented in the manuscript. The results and analysis are clearly presented and illustrate both an exceptional level of control that is possible in their gyro system as well as an excellent understanding of the coupling mechanisms. The authors have also given extensive background in the way of citations to the main elements of their work.

A question for publication in Nature Communications is to ask what elements of the work represent new results (conceptual, technique-driven or actual new phenomena) versus results that are significant improvements in observation and control enabled by the device platform. For this question, I am not convinced that the present work (while elegant and beautifully presented/executed) presents new techniques or has measured new phenomena. Perhaps the authors need to be clearer on this point in their paper, however, at present it is difficult to perceive an advance that justifies publication in Nature Communications. For example, my sense from the paper is that the techniques applied (tuning control and parametric nonlinear coupling) have been previously reported. Are there techniques or results that are significant and entirely new?

A related point noted by the authors is the possible connection of phenomena studied here to measurable improvements in the gyro performance. For example, the authors wrote in the discussion and conclusion, ``More interestingly, dynamical sideband coupling may be very useful in terms of enhancing sensor performance...'. Could the authors follow-up on this point to show experimentally how the strong dynamical coupling can improve the gyro performance? Such a measurement that connects the exceptional control demonstrated in the paper to device performance would be of great interest to both mechanics and gyro communities. If the authors can do this, then I would support publication.

Response to Reviewers

We thank the reviewers for their detailed reviews. The manuscript has been modified to address the feedback received. Please find below our response to the reviewers. Reviewers' comments are marked in **red with sans-serif font**, authors' responses are in **black with serif font**, and the **blue parts with serif font** indicate the corresponding modifications in the paper and supplementary information.

Reviewers' comments:

Reviewer #1 (Remarks to the Author):

The manuscript describes the tuning of mode coupling in a MEMS resonator through electrostatic control. A number of different coupling behaviors are observed and carefully analyzed, and the analytical results beautifully match the experimental data. The extent of the work and the quality of results are of great potential to make a good publication; however, a number of issues should be addressed before this manuscript is fully suitable for publication in Nature Communications.

The authors mentioned a number of tunable mode coupling work. It would be interesting to compare the electrostatic-tuned mechanism reported in this manuscript with, say, tension-induced ones, such as those in references 27, 29, 30, and Nano Lett. 15, 6727 (2015), and articulate the unique advantages of electrostatic tuning demonstrated in this work.

We would like to express our sincere thanks to Reviewer #1 for the comprehensive comments and valuable suggestions, which have greatly enhanced the paper. The tension-induced mechanism was usually used for the clamped-clamped designed resonators or those that are made by thin-film or low-dimensional materials, because it is relatively easier to generate tension in these resonators. However, for more complex resonator topologies (such as center-anchored low-stress disk or ring resonators) or resonators made by bulk micromachining technologies (such as widely used high-aspect-ratio silicon resonators), the efficiency of tension-induced pumping would be degraded. The electrostatic tuning mechanism has no such limits. It is thus applicable to a wider range of resonator topologies and fabrication technologies, though the resonator body should be sufficiently conductive.

As Reviewer #1 suggested, we have strengthened the presentation in Line 396-407, the first paragraph of Section III Discussion as follows: “Electrostatic pump based manipulations show consequent advantages with respect to tension-mediated manipulations. Depending on the topology of the resonators, the effect of tension-induced manipulations can vary. For instance, its effect can be significant for clamped-clamped or thin film resonators, but negligible for centrally anchored or bulk- fabrication-process resonators, such as the ring resonator in this work, if the same tension is applied. In comparison, the electrostatic-mediated

manipulating approach removes such topology restrictions for capacitively transduced resonators, hence making it applicable for a wider range of resonator designs.”

The authors showed that the modes II-1 and II-2 can be measured and tuned using different electrodes. Here, it was determined that the two intrinsic II modes (without any tuning), whose frequencies split due to asymmetries due to fabrication process, have mode shapes that are 18 degrees off from the x and y axis in the setup coordinate. Why didn't the authors choose to adjust their measurement arrangement (by connecting to different electrodes) so that the electrodes are best aligned with the intrinsic mode shapes? Since the authors have electrode every 22.5 degrees, it seems natural to rotate all the connections by one electrode so that misalignment between drive/probe and the intrinsic mode shape is off by only 4.5 degrees.

We are really sorry for the unclear presentation of the coordinate system for this ring resonator. It is known that for in-plane motion of flexural ring structures, the principal axes of each pair of order- n degenerate wineglass normal modes have an angular interval of $90^\circ/n$ in real setup coordinates. In this research, order-2 degenerate modes have an azimuthal angular interval of 45° . The order-2 modes are equivalent to reduced-order two-degree-of-freedom lumped parameter model in Cartesian coordinates, in which the two equivalent modes are orthogonal to each other with azimuthal angular interval of 90° . This equivalence is revealed in the newly added Figure 2(a,b). It can be seen that the angle in this equivalent system (Cartesian coordinates) is double of that in the real system (setup coordinates).

The equations of motion are based on the equivalent coordinates. The initial misalignment angle θ is estimated to be 18° in the equivalent coordinates. In fact, the initial is 9° in real setup coordinates, in which electrodes are separated by 22.5° . Thus, if we rotate all the connections by one electrode, the real misalignment angle would be 13.5° , which is larger than the current 9° .

We apologize for the ambiguous description in the previous manuscript. In order to avoid the confusion, we only talk about angles in equivalent coordinates. The electrodes that were distinguished by angles in the original paper are now marked with numbers, as shown in line 124-134, the second paragraph of Subsection A of Section II and Figure 1(a) of the revised paper.

In Section I, Subsection B, Part 1, line 138-148 and line 165-166 of the revised paper, we clearly elaborate upon this equivalence.

Moreover, we add a description about the equivalent two-degree-of-freedom lumped parameter system in Supplementary Information Section I, paragraph 3.

It would be helpful to label some of the angles in Fig. 1 so the readers can more easily relate to the description in the text. Since in the actual measurements the authors choose to use the measurement arrangement as it is, it may be more intuitive to show the mode shapes with their actual orientation in Fig. 1b (with theta also shown), which will help the readers better understand the part of the misalignment between the mode shape and the electrodes.

We really appreciate this useful suggestion. We have made modifications based on this comment. In the previous manuscript, the electrodes are distinguished by its angle in the setup coordinates. This may have caused confusion because the angle in setup coordinates is the half of that in equivalent system. Since the theoretical model is based on the equivalent system, we try to avoid using the setup coordinates in the revised paper. Thus, the electrodes in Figure 1(a) are labeled with numbers instead of angles. We have also adjusted the electrode description in the end of Section II, Subsection A, which are marked with red.

We have added schematic mode shape of the order-2 modes with the misalignment angle in Figure 2(a). Its equivalent system is also depicted in Figure 2(b). The initial misalignment condition is revealed in Figure 2(a),(b). In Figure 1(b), we only intend to provide the mode information (eigenfrequencies and mode shapes) corresponding to the device description part of Section II, Subsection A.

The authors used V_{t1} and V_{t2} to tune the frequencies of the II-1 and II-2 modes. Since these two mode shapes are 45 degrees from each other in real space, it would be intuitive to use two electrodes that are also 45 degrees apart to tune these two modes (as the respective nodes and antinodes in the two modes are also 45 degrees apart). Why do the authors choose to use two electrodes that are only 22.5 degrees apart? It seems a bit counterintuitive—suppose one electrode tunes one mode most effectively, then the other electrode would be sitting almost in between the two mode shapes, and would likely be tuning both II modes together?

We thank Reviewer #1 for raising this great comment. Effectively modeling the electrostatic tuning process based on linear transformation is one of the advances that we make in this paper. In order to make the tuning process clearer, we have added the influences of V_{t1} and V_{t2} on the eigenfrequencies modes II-1 and II-2 in the end of Section II of the Supplementary Information, which are marked in red.

In fact, the parameters of significance in the tuning process are the frequency difference $\Delta\omega$ of the order-2 normal modes and the misalignment angle θ . The reason we apply an in-axis tuning (V_{t1}) and an off-axis tuning (V_{t2}) is that this configuration can separately allow tuning of $\Delta\omega$ and θ . For instance, if θ is near 0° , it can be seen from equation (S12) that, θ is sensitive to off-axis tuning (Δ_{t2}) whereas insensitive to the in-axis tuning (Δ_{t1}). In this condition, one can also show that $\Delta\omega$ is sensitive to in-axis tuning (Δ_{t1}) whereas insensitive to the off-axis tuning (Δ_{t2}). On the other hand, if θ is near 45° , θ is sensitive to in-axis tuning (Δ_{t1}) whereas insensitive to the off-axis tuning (Δ_{t2}), and $\Delta\omega$ is sensitive to off-axis tuning (Δ_{t2}) whereas insensitive to the in-axis tuning (Δ_{t1}). Subject to the initial misalignment, a specific group of $\Delta\omega$ and θ values can be obtained by appropriately choosing V_{t1} and V_{t2} .

However, if we apply tuning voltages V_{t1} on x direction and V_{t2} on y direction, which means Δ_{t1} and Δ_{t2} are both in-axis, the effect of Δ_{t2} on $\Delta\omega$ or θ is equivalent to that of $-\Delta_{t1}$. In this case, we cannot individually change $\Delta\omega$ and θ .

I do notice that tuning using V_{t1} (with measurement both along x and y axes are shown in Fig. 2b and S1c), in which clearly one mode is being effectively tuned from 134.0 to 134.4 kHz, while the other mode is mostly intact. I

wonder what happens when the similar experiment is repeated for V_{t2} —do both modes move together? Or away from each other? Since it would be unlikely that V_{t2} can effectively tune just one mode and leave the other one intact due to the reason in the previous paragraph.

Changing the eigenfrequency of one mode while keeping that of the other mode intact is not objective of this work. Our goal in this paper is to separately change $\Delta\omega$ and θ . In fact, the best way to separately change ω_{II-1} and ω_{II-2} is to first adjust θ to 0° by applying an off-axis tuning, and then independently tune II-1 or II-2 using electrodes placed along the x or y axis. When θ is 0° , the principal axes are perfectly aligned with x - o - y coordinates, which means that states probed from x - o - y would be the order-2 normal modes themselves. In this case, no avoided crossing phenomenon would be observed if no other coupling mechanism is considered. Though this tuning process is very interesting, it is out of the scope of this paper. We think it would be better to discuss it in our more relevant future works instead of this paper.

The effects on eigenfrequencies and θ of the off-axis tuning V_{t2} are explicitly presented in equations (S12) and (S15) of the revised Supplementary Information. When only the off-axis tuning V_{t2} is applied, the simulated frequency responses detected along x and y axes at ω_d with different values of tuning voltages are shown in the following Figure R1.

Fig. R1. Simulated frequency responses detected along x (a) and y (b) axes at ω_d with different values of off-axis tuning voltages V_{t2} .

In this paper, off-axis tuning V_{t2} and in-axis tuning V_{t1} are used together to separately control $\Delta\omega$ and θ . The individual off-axis tuning process is not of particular interest or significance in this work. Thus, we have provided the simulation result here and not included it in the paper, and will discuss it in detail in a more relevant future work.

Meanwhile, it would be intuitive to illustrate this process (tuning using V_{t1}) using similar presentation as in Fig. 1b to supplement the data in Fig. 2b, by showing the frequency response curve AND mode shapes as V_{t1} increases, so the readers can better understand how II-1 and II-2 gradually evolve into H-1 and H-2; how does the mode shape of H-1 and H-2 look like; what does the mode shape look like during anti-crossing, etc. Therefore I invite the authors to seriously consider moving Fig. 2a,b,c into a new figure, and adding a few line plots of frequency response with simulated mode shape illustrations as in Fig. 1b, with corresponding V_{t1} values outlined on the color plot (current Fig. 2b), so help readers comprehend this process better.

We really appreciate this excellent suggestion. We have revised the paper based on this comment. Please see Figure 2 of the revised paper, which has covered all requirements above. The frequency response curves with different tuning voltages and mode shapes with corresponding misalignment angles are additionally presented in Figure 2(f).

After equation 2, the authors mentioned that “The amplitude of the first-order term is $2V_0/V_p$ times that of the second-order term”. Please double check the math and see if it would be 4 instead of 2.

We thank Reviewer #1 for carefully reading our paper and correcting this error. This should be $4V_0/V_p$ instead of $2V_0/V_p$. We have also checked the mathematics of the whole paper, and no such mistake was found elsewhere.

For the application of the pump signal, there is a lot information shown in Fig. 2d-i, and in the current presentation it is rather dense and a bit challenging to follow for most readers. I suggest the authors considering making it easier for the average reader by doing the following:

- Make Fig. 2d-i a separate figure of its own. Actually with Fig. 2a-c moved out this should have been achieved already.
- Align Fig. 2d with 2f,g vertically on the frequency axis. The authors might consider reduce the span of the low frequency part a bit (to the left of the axis break), and make the high frequency part (to the right of the axis break) exactly aligned with the color plots.
- Use thin vertical dashed lines through 2d,f,g to indicate the drive, idler, and sidebands (1-4).
- Add a horizontal dashed line in 2f to indicate $\omega_p = \Delta\omega$, so that the readers can see which horizontal slice in fig. 2f corresponds to Fig. 2d.
- Similarly align Fig. 2e with Fig. 2f and 2g (maybe from underneath, or in between the two, depending on which arrangement is best for presentation). Draw similar vertical dashed lines for bands 5-14 in Fig. 2e.
- Add a horizontal dashed line in 2f or 2g to indicate $\omega_p = 1/2 \Delta\omega$, so that the readers can see which horizontal slice in fig. 2f or 2g corresponds to Fig. 2e.
- Use horizontal dashed lines in Fig. 2f and/or 2g to indicate where the pump strength sweep in Fig. 2h,i is conducted.

Taking these measures can help better illustrate the experiment process, and allow the readers to quickly understand the experiment data.

We really thank Reviewer #1 for enthusiastically providing such thoughtful and significant suggestions. We agree that the original presentation in this figure might be too dense. Thus, we have substantially revised it. We have removed the content on mechanical hybrid coupling. However, we did not vertically align Figure 3(a) and (b) with Figure 3(c) and (d) based on the following reason. Figure 3(a) and (b) are schematic spectra, the abscissa of which represents the *general* frequency. Whereas Figure 3(c) and (d) are frequency responses, the abscissa of which represents the *drive* (*probe*) frequency. Figure 3(a) and (b) are not slices of Figure 3(c) or (d). In this work, frequency responses are obtained by sweeping the drive frequency ω_d and probing its

response displacement amplitudes demodulated by the reference signal with frequency of ω_d . Thus, the responses in Figure 3(c) and (d) are the displacements driven by drive tone and sidebands at ω_d . Sidebands with other frequencies $\omega_{\text{other}} \neq \omega_d$ in Figure 3(a) and (b) will be not shown in Figure 3(c) or (d), instead, should be depicted in responses demodulated by reference signal with frequency of ω_{other} . In order to avoid confusion, we have deleted the initial frequency response curves in the original schematic spectra, which were only used for noting the resonant frequencies.

In some cases, sidebands in the schematic spectrum may coincide with resonance peaks in frequency response. However, in this study, we find that resonance peaks may not perfectly aligned with sidebands. The avoided crossing induced by the very strong first-order dynamical coupling will shift the lower branch of resonance peak in Figure 3(c) and (d). This phenomenon will also make the second-order avoided crossings take places at a pump frequency higher than $\Delta\omega/2$, as shown in Figure 3(c) and (d). However, this definitely does not mean that the second-order dynamical coupling takes place at pump frequency higher than $\Delta\omega/2$, The second-order dynamical coupling still takes place at pump frequency of exactly $\Delta\omega/2$, but the observed second-order avoided crossing is simultaneously affected by both the first- and the second-order coupling.

We do provide some response slices in Figure 3(e) to support the avoided-crossing evolution process. In order to better illustrate all the peaks, we chose the logarithmic form in the amplitude axis. We have placed this selected frequency response curves at the right side of Figure 3(c) and (d) to avoid the whole figure from being too protracted.

Horizontal dashed lines are added in Figure 3(c) and (d) corresponding to different response curves in Figure 3(e). Vertical dashed lines are added in Figure 3(c) and (d) to indicate the resonance frequencies of the modes II-1 and II-2 after a 3 V pump is applied.

Horizontal dashed lines are added in Figure 3(g) and (h) to indicate the pump strength of the dynamical experiments conducted in Figure 3(c-e).

It should be noted that, the DC term in the pump Δ_p (equation (2)) will affect the resonant frequencies and misalignment θ based on the electrostatic tuning theory in *Section II of the Supplementary Information*. Thus, resonant frequencies $\omega_{\text{II-1}}$ and $\omega_{\text{II-2}}$ are not constants in Figure 2(g,h) when pump strength V_p changes.

The related revisions include the modified Figure 3 and line 246-260, the third paragraph in the left column of page 5.

When discussing the nonlinearity in the resonator, the authors mentioned that “It is also known that the doping process impacts material nonlinearity [51], which is often dominant in bulk-mode resonators. However, flexural resonators such as the ring resonator in this study [51] are limited by nonlinear effects due to the imposed electrostatic field.” Why material nonlinearity is not important in this resonator under study? It is unclear from the writing.

In Fig. 3b,c, why is the legend arranged with 2mV on top and 10mV at bottom? That arrangement seems counterintuitive and completely opposite

from that of the actual data (making those print on black-and-white really hard to follow)

We greatly thank Reviewer 1# for pointing out this problem. It is true that there are many nonlinear mechanisms can be potentially operative in this resonator, including mechanical nonlinearity, material nonlinearity, and the electrostatic nonlinearity. We have verified that this resonator is dominated by electrostatic nonlinearity by additionally implementing further experiments (Shown in the newly added Subsection A, Section IV of the revised Supplementary Information), in which the DC voltage is changed while the product of the DC voltage and AC drive voltage amplitude (proportional to actuation force) is kept constant. The electrostatic nonlinearity is sensitive to the DC voltage whereas other nonlinearities are not. We observe that the resonator shows stiffness-softening Duffing nonlinear frequency responses, and the nonlinearity strength is highly DC voltage-dependent (Fig. S7 and S8 of the revised Supplementary Information). Thus, we can draw a conclusion that the ring resonator used in this paper is dominated by electrostatic nonlinearity in the setup of this paper, in which DC voltage V_0 is set to be 30 V.

The resonator structure is depicted in the newly added Figure S1b of the supplementary Information. The ring with width of 3 μm is very thin compared to the overall dimension of 720 μm . The ring resonator in this study is a typical flexural mode resonator, whereas material nonlinearity is typically evident in bulk resonators [52]. Moreover, this resonator is clamped at the center. The clamping forces are perpendicular to the mid-planes of the rings. Thus, it would be relatively hard to generate tension in the flexural rings. Another important feature of this ring resonator is the very small capacitive gap (1.5 μm). The dominance of the electrostatic nonlinearity is consistent with physical intuition and previous work on similar systems.

In this study, we demonstrate that electrostatic field can also provide very strong nonlinear mode interactions in single resonator. The observed experimental nonlinear coupling phenomena matches very well with the electrostatic-theory-based model. The mode interaction caused by material nonlinearity is also a very interesting topic, which deserves a future study using other bulk-mode resonators dominated by material nonlinearity.

Based on this comment, we have revised the nonlinearity discussion part at the beginning of Subsection C, Section II. Moreover, we have removed the original Figure 3 (a)(b), in which the AC drive voltages are increased while the DC voltage is kept unchanged, because they do not provide much more helpful information. Instead, we have provided the verification experiment of changing DC while keep DC \times AC unchanged in Subsection A, Section IV and Figures S8 and S9 of the revised Supplementary Information.

In Fig. 3d,e color plots, it would be helpful to label II-1 and II-2 modes. For the experiment in Fig. 3f, it would be helpful to use vertical dashed lines to indicate II-1, II-2, and a sloped dashed line to show (III-1)-(ω_p), so the readers to relate to the features on the figure. If possible, it would also be helpful to show the detected signal for mode III-1 at its frequency when

sweeping ω_p as in Fig. 3f, to show the power is being transferred from the II modes to the III-1 mode.

We thank Reviewer #1 for giving these suggestions. As the Reviewer suggested, we have labeled the modes in the frequency responses of revised Figure 4(d,e) (the original Figure 3(d,e)). In Figure 4(f,g), we have also added vertical dashed lines to indicate II-1 and II-2. Slope dashed lines of $\omega_{III-1} - \omega_p$ are also plotted. Reviewer #1 suggests detecting the displacement of mode III-1 while driving ω_d at and pumping at ω_p to show the power transfer, needed to demodulate the response using reference with frequency of $\omega_d + \omega_p$. Directly observing the energy transfer is one of our primary goals. However, we have some hardware problems in realizing this that can hardly be solved in a short time. Moreover, we think that the observed avoided crossings in Figure 4(f,g) is sufficient for demonstrating the mode interactions in this study. Considering the length of this paper, we hope Reviewer could allow us to implement this experiment in another future work.

Similar to the II case, it would be helpful to use horizontal dashed line in Fig. 3f to indicate where the experiments in fig. 3h,j are conducted.

We have added horizontal dashed line in Figure 4(f) and Figure 4(h,j) to indicate the same frequency responses, which are labeled as lines vi and vii.

Also what's the reason for the apparent splitting observed in 3h and 3j? It was clear from the current writing. I am not sure if the nonlinearity shown in Fig. 3b-e is necessary for observing the mode coupling in Fig. 3f-k. From the color scale in Fig. 2f it seems the authors are actuating II-1 only using 2mV (the smallest one in Fig. 2c), and still clearly observed the mode coupling. Then one would conclude the nonlinearity shown in Fig. 2b,c are not necessary for observing the mode coupling in Fig. 3f-k. In that case, maybe the authors can re-arrange the order and tell the story of Fig. 3f-k first? As that part is more in the same line with the story in the previous figure. And Fig. 3b-e may be moved to the end as a separate side story, or into the SI, as it does not affect the main story line of the main text.

We are very sorry for not making our point clear enough. The first thing that we want to illuminate is that two mechanical modes in one resonator can interact with each other if they share a common biased capacitor, which has not been perceived before. We realize that we have not described this coupling mechanism clearly, Reviewer #2 also pointed out that we need to present our physical consideration more clearly. The physical picture of the electrostatic parametric coupling mechanism is as follows: When two modes II and III are simultaneously actuated, their displacements are superposed (Figure 4(c)). They share common capacitors, and both of them will affect the capacitance gap. The affected gap is $d_0 + x_{II} + x_{III}$, where d_0 is the initial gap. If a bias voltage ΔV is applied between the electrode and the resonator body, electrostatic field will introduce a modulation to the effective stiffness of mode II,

$\frac{A_c \epsilon_0 \Delta V^2}{m_{II} (d_0 + x_{II} + x_{III})^3}$, where A_c , ϵ_0 , m_{II} are the capacitance area, permittivity of vacuum,

and effective mass of mode II, respectively. Mode II is actuated into linear region, and the effect of the mode II displacement to gap can be neglected. However, mode III is

actuated into strong nonlinear condition, which will greatly affect the capacitance gap, and then remarkably modulate the stiffness of mode II stiffness. To sum up, the displacement of mode III can modify the resonant frequency of mode II, just like the tension-induced parametric coupling reported in PRL 105, 117205 (2010). Furthermore, the electrostatic potential may also introduce other linear, quadratic and cubic coupling terms. A more systematic description is provided in Supplementary Information Section IV.

In this research, the stiffness-softening electrostatic nonlinearity is dominant. The most important feature that can be used to tell that the parametric coupling comes from the electrostatic field is that the actuation of mode III will shift the frequency of mode II to lower values (Figure 4(d,e)), which is opposite to that of the mechanical nonlinear parametric coupling.

We have enhanced the electrostatic nonlinear coupling mechanism description in paragraphs 2, 4, 5 of the Part I, Subsection C Section II of the revised paper.

The authors claimed that “In this case, hybrid states H-1, H-2, and mode III-1 construct a coupled-three-mode configuration”. However, in the experiment shown in Fig. 3f, by sweeping ω_p mode III-1 is sequentially coupled to II-1 and II-2, but the responses of II-1 and II-2 never cross each other. So it appears what the authors have demonstrated are two separate two-mode couplings II-1 to III-1, and II-2 to III-1, in the same measurement, which can hardly be called a “coupled-three-mode”—because if one extends the range of ω_p a bit more III-2 mode would also be involved—would one call it “coupled-four-mode” then (and one would imagine the data in 3f would look like a skewed “#”)? There is no limit to how many mode coupling one could observe in such experiments, but each crossing (or avoided crossing) represents the coupling between two individual modes—unless one tune the frequency and make 3 distinct modes cross at the same point. In fact, the authors may consider including data in Fig. 3f with ω_p go from below III-1 to above III-2 to show additional couplings, which would be more illustrative.

We thank the reviewer for this insightful comment and agree with this viewpoint. In this work, we did not intend to demonstrate the simultaneous dynamical coupling of H-1 to III-1 and H-2 to III-1, or the so-called three-mode coupling [Applied Physics Letters 108, 227402 (2016)]. What we want to demonstrate is that this electrostatically driven system is abundant in mode coupling phenomena. We avoided using the expression “three-mode coupling system”. However, the term of “coupled-three-mode” is still misleading. Thus, in the revised paper, we describe this as “coupling-abundant multiple-mode system”, which we believe is more suitable.

In fact, H-1 and H-2 are coupled hybrid states, this coupling can also be revealed by Figure 4(h) and (j). In Figure 4(h), mode II-1 splits into two peaks due to dynamical coupling of H-1 and III-1. An avoided crossing can be observed between the higher splitting branch of II-1 and mode II-2, which is produced by the hybrid coupling of H-1 and H-2. Likewise, in Figure 4(j), mode II-2 splits into two peaks due to dynamical coupling of H-2 and III-1. The avoided crossing between the lower

splitting branch of II-2 and mode II-1 is also produced by the hybrid coupling of H-1 and H-2.

We agree that the numbers of mode coupling that can be observed in this system are not limited to those shown here. Nevertheless, we do think that it is of significance to find a system that can easily provide an abundance of mode coupling phenomena, because a priori engineering of such systems is not trivial. In this work, we demonstrate that electrostatically transduced mechanical systems have the potential to couple multiple modes, and more importantly, we provide a coupling-abundant multiple-mode platform for further mode coupling research.

Finding numerous distinct mode coupling is the basis for realizing simultaneous multi-mode coupling. One of our future research directions is to use this coupling-abundant multiple-mode system to simultaneously couple different modes, and study more interesting multiple-mode coupling phenomena.

It is true that if the pump is swept from below III-1 to above III-2, a skewed “#” would be observed. In fact, we have separately demonstrated the coupling of III-1 to II-1 and II-2 (Figure 3(f,g) of the original paper), and coupling of III-2 to II-1 and II-2 (Figure S7 of the original Supplementary Information). We have merged those parts into the revised Figure 4(f), which shows a broken-axis skewed “#” configuration.

A number of typos should also be fixed before resubmission, such as “annihilation”. It would be helpful to show the detailed structure (such as SEM images) of the resonator in SI.

We really appreciate for pointing out these typos, which have been corrected in the revised paper. As reviewer suggested, we have added more detailed description of the ring resonator in the Supplementary Information Section I, including a cross-section schematic in the new Figure S1.

Overall this is a solid piece of work, and a number of interesting observations are being made with good match between numerical model and experiment data. While the current presentation is a bit too dense and difficult to follow by average readers, after appropriate and sufficient improvement I will re-consider recommendation of its publication.

Once again, we sincerely thank Reviewer #1 for giving such excellent comments and suggestions, which are really helpful and valuable for improving this paper. We have carefully studied and responded all the comments, and made considerable modifications based on those suggestions, which we hope meet with approval.

Reviewer #2 (Remarks to the Author):

The manuscript has investigated sideband coupling in a silicon-based gyroscopic ring resonator. Parametric coupling is demonstrated in both two-mode and three-mode configurations. Overall, the paper is well-written and well-organized. However, I cannot recommend its publication in Nature Communications at present based on the following comments:

1. I'm wondering what is the difference between this paper and other works on parametric coupling in micro-beam or micro-cantilever based mechanical resonators. For example, H. Okamoto et al. have studied dynamic

coupling and observed high-order process under similar periodic pumping set-up (see Nature Physics 9, 480 (2013)). Can the authors further illustrate the novelty of this manuscript?

We sincerely thank Reviewer #2 for the valuable comments. The revision guided by these comments has greatly enhanced our paper.

In this paper, we demonstrate dynamically tuning certain intrinsic modal interactions into tunable dynamical coupling based on an electrostatic parametric pump in a capacitive silicon micro-electro-mechanical resonator. The dynamical manipulation between an optical/microwave cavity and a mechanical mode is well studied in electro-optic hybrid systems [Science 321, 1172, (2008)]. The first demonstration of dynamical manipulation between two coupled mechanical modes based on piezoelectric parametric pump was proposed by Yamaguchi group [Nature Physics 8, 387–392 (2012); Nature Physics 9, 480–484 (2013)], which has also inspired further works [Nano Letters 15, 6727 (2015); Nature Nanotechnology 11, 741 (2016); Nature Nanotechnology 11, 747 (2016); etc.]

However, this work is the first to demonstrate a testbed based on nonlinear electrostatic parametric coupling in a single resonator. This is a highly versatile platform for exploring mode coupling as is demonstrated through both theory and experiment. As opposed to previous works, the degree of tunability of the coupling behavior is significant and this approach can be extended to a wide variety of electrostatically-transduced MEMS/NEMS of practical import (e.g. gyroscopes or mode-localized sensors). Further, our work clearly describes the physics of associated intrinsic mode coupling mechanisms as opposed to previous works which simply reported the experimental observations and employed generic models to model the dynamics. A further advance is that we introduced new coupling manipulation techniques. Thus, this device is demonstrated to as a coupling-abundant multiple mode system, which is difficult to construct in a single device by a-priori engineering of the resonator.

Note that, in previous works, the only perceived nonlinear *parametric* mode coupling mechanism is due to the mechanical tension-induced nonlinearity [Physical Review Letters 105, 117205 (2010)], and most mode coupling manipulations are based on this understanding [Nature Physics 8, 387–392 (2012); Nature Physics 9, 480–484 (2013); Nano Letters 13, 1622–1626 (2013); Nano Letters 15, 6727 (2015); Nature Nanotechnology 11 (2016); Nature Nanotechnology 11, 747 (2016); etc.]

This includes previous works that employ electrostatic transduction. In this work, we demonstrate that, electrostatic field can also cause nonlinear *parametric* coupling among mechanical modes, which acts in the opposite sense to tension-induced nonlinear *parametric* coupling. Another advance is that, though electrostatic coupling has been extensively researched [Journal of Applied Physics 114, 4469 (2013); Journal of Microelectromechanical Systems 11, 802–807 (2002); *Nonlinear Dynamics of Nanomechanical and Micromechanical Resonators*, WileyVCH Verlag GmbH & Co. KGaA, 2008.], electrostatic *nonlinear* coupling among different modes in *one* resonator has never been elaborated to this level of detail.

Apart from the potential immediate practical relevance of this work to MEMS gyroscopes and other resonant sensors, this work could allow more flexible testbeds for fundamental research on mode coupling.

Based on this comment, we have revised the last paragraph of the Introduction Section and the first paragraph of the Discussion and conclusion Section to better illustrate the novelty of this work.

2. The authors claim that there is an initial misalignment theta, induced by manufacturing process. Also, under some particular value, for example theta=45 degree, the coupling pattern is changed (Figure S2). Is it possible to tune the pattern in Figure 2f to that in Figure S2c to cancel second-order coupling, since theta can be tuned electrically by changing Vt1 and Vt2?

We thank Reviewer #2 for pointing out this interesting issue. It is possible to adjust θ to cancel the second-order dynamical coupling. We have added the second-order dynamical coupling strength in equation (S45) of the revised Supplementary Information, which is given by

$$g_2 = \frac{2\kappa^2 V_0^2 V_p^2 \sin(2\theta) \left(\frac{\cos^2 \theta}{\omega_{\text{II-1}} + 3\omega_{\text{II-2}}} - \frac{\sin^2 \theta}{3\omega_{\text{II-1}} + \omega_{\text{II-2}}} \right)}{2(\omega_{\text{II-2}} - \omega_{\text{II-1}}) \sqrt{\omega_{\text{II-1}} \omega_{\text{II-2}}}}$$

The second-order coupling would be canceled if the misalignment is tuned to be $\theta = \arctan \sqrt{(3\omega_{\text{II-1}} + \omega_{\text{II-2}}) / (\omega_{\text{II-1}} + 3\omega_{\text{II-2}})}$. In this work, the difference between $\omega_{\text{II-1}}$ and $\omega_{\text{II-2}}$ is much smaller than $\omega_{\text{II-1}}$ and $\omega_{\text{II-2}}$ themselves, and $3\omega_{\text{II-1}} + \omega_{\text{II-2}} \approx \omega_{\text{II-1}} + 3\omega_{\text{II-2}}$. Therefore $\theta \approx 45$ will cancel the second-order dynamical coupling. The misalignment tuning process is described by equation (S12) of the revised Supplementary Information.

We have enhanced this issue in line 235-241, the second paragraph, left column, page 5 of the revised paper. In line 278-281 of page 5, we also included a statement that adjusting θ provides a new degree of freedom to control dynamical couplings.

3. About the coupling mechanism in Figure 3a, what is the shared capacitance? Is it a real capacitance (or sum of some real capacitances) or an imaginary effective capacitance? What is the physical consideration behind this model?

We are very sorry for the previous unclear description about capacitive parametric coupling. The capacitance is real, which is the sum of all the polarized capacitances. The physical picture of the electrostatic parametric coupling mechanism is as follows: When two modes II and III are simultaneously actuated, their displacements are superposed (Figure 4(a)). They share a common capacitor, and both of them will affect the capacitance gap. The affected gap is $d_0 + x_{\text{II}} + x_{\text{III}}$, where d_0 is the initial gap. If a bias voltage ΔV is applied between the electrode and the resonator body, electrostatic field will introduce a modulation on the effective stiffness of mode

II, $\frac{A_c \epsilon_0 \Delta V^2}{m_{\text{II}} (d_0 + x_{\text{II}} + x_{\text{III}})^3}$, where A_c , ϵ_0 , m_{II} are the capacitance area, permittivity of vacuum,

and effective mass of mode II, respectively. Mode III is actuated into a strong nonlinear condition, which will vary the capacitance gap, and then modulate the stiffness of mode II. To sum up, the displacement of mode III can modify the resonant frequency of mode II, just like the tension-induced parametric coupling reported in

PRL 105, 117205 (2010). Furthermore, the electrostatic potential may also introduce other linear, quadratic and cubic coupling terms. A more systematic description is provided in Subsection B, Section IV of the Supplementary Information.

In order to better illustrate the capacitive coupling of two modes, we have replaced the original Figure 3(a) diagram by the revised Figure 4(a-c), in which the displacement superposition is demonstrated by a transient sketch of the simultaneously actuated modes. The physical description is enhanced in paragraph 4 of the Part I, Subsection C, Section II. The theoretical model of this coupling is provided in detail in Subsection B, Section IV of the Supplementary Information to support the physical description.

4. In Figure 2h, the frequency response seems to be asymmetric when pump strength is larger than 6V, after two first-order response crossing each other. Similarly, the frequency response of mode II-1 becomes asymmetric from the very beginning at around pump strength=1V in Figure 3h. Any explanations about this?

The asymmetries are caused by the DC term in the pump (equation (2)). This DC term is applied along y direction, which can affect the misalignment and resonant frequencies. We repeated the sweeps in Figure 3(f) and applied *off-resonance* pumps with different pump strength. In this condition only the DC term in the pump signal would affect the system. The effect of DC term on the order-2 modes is provided in Figure S6 of the revised Supplementary Information. The theoretical result coincides well with the experimental data. We have included this in the revised Figure 3(f,g), which perfectly coincides with the overall trend of the responses. The frequency response of mode II-1 becomes asymmetric from the very beginning as this DC term can affect the misalignment angle θ . In the simulation (Fig. 3(g)), we have included the influence of this DC term. If the DC term is removed, the simulation result would be different (as noted in the following Figure R2a). If the DC term is considered, the simulation result would be as seen in Figure R2b, which coincides with Figure 3(g) of the paper. Thus, it can be seen that, if the DC term is removed, the frequency response of mode II-1 will not be asymmetric from the very beginning.

Fig. R2. Simulated frequency responses as the functions of V_p . The DC term is removed in (a) while considered in (b).

In line 182-188, paragraph 2 of the Part 2, Subsection B, Section II and in line 284-286, the last paragraph of the Part 2, Subsection B, Section II, we have made a statement about the influence of the DC term.

We have explained the influence of this DC term in detail based on the electrostatic tuning theory in Subsection D, Section III of the Supplementary Information.

5. Several misspelling: for example anihilation in Page4, congurtion in captions of Figure 3.

We really thank Reviewer #2 for reading our paper so carefully. The misspellings have been corrected. We have checked the writing of the whole paper to avoid similar mistakes.

Overall, I cannot recommend publication of this manuscript before all the comments being addressed.

We sincerely thank Reviewer #2 for those constructive comments, all of which have been addressed. The revisions based on the comments have strengthened our paper from the aspects of significance and stringency. We hope our revisions and responses meet the expectations of the reviewer.

Reviewer #3 (Remarks to the Author):

This manuscript investigates modal coupling in microelectromechanical gyroscopic ring resonators. The authors demonstrate precise control the mode coupling by observation of avoided mode crossings under dynamic capacitive tuning. In addition, they studied dynamical coupling effects using a nonlinear parametric coupling. Detailed analyses, simulations, and experiments were presented in the manuscript. The results and analysis are clearly presented and illustrate both an exceptional level of control that is possible in their gyro system as well as an excellent understanding of the coupling mechanisms. The authors have also given extensive background in the way of citations to the main elements of their work.

A question for publication in Nature Communications is to ask what elements of the work represent new results (conceptual, technique-driven or actual new phenomena) versus results that are significant improvements in observation and control enabled by the device platform. For this question, I am not convinced that the present work (while elegant and beautifully presented/executed) presents new techniques or has measured new phenomena. Perhaps the authors need to be clearer on this point in their paper, however, at present it is difficult to perceive an advance that justifies publication in Nature Communications. For example, my sense from the paper is that the techniques applied (tuning control and parametric nonlinear coupling) have been previously reported. Are there techniques or results that are significant and entirely new?

We sincerely thank Reviewer #3 for reviewing our paper so carefully. We apologize for not having expressed the novelty and significance of our work in a clearer manner. We want to present the novelty claim here again, for the convenience of Reviewer #3. We have considered the comments of both Reviewer #2 and Reviewer #3 when making this explanation.

In this paper, we demonstrate dynamically tuning certain intrinsic modal interactions into tunable dynamical coupling based on an electrostatic parametric pump in a capacitive silicon micro-electro-mechanical resonator. The dynamical manipulation between an optical/microwave cavity and a mechanical mode is well studied in electro-optic hybrid systems [Science 321, 1172, (2008)]. The first demonstration of dynamical manipulation between two coupled mechanical modes based on piezoelectric parametric pump was proposed by Yamaguchi group [Nature Physics 8, 387–392 (2012); Nature Physics 9, 480–484 (2013)], which has also inspired further works [Nano Letters 15, 6727 (2015); Nature Nanotechnology 11, 741 (2016); Nature Nanotechnology 11, 747 (2016); etc.]

However, this work is the first to demonstrate a testbed based on nonlinear electrostatic parametric coupling in a single resonator. This is a highly versatile platform for exploring mode coupling as is demonstrated through both theory and experiment. As opposed to previous works, the degree of tunability of the coupling behavior is significant and this approach can be extended to a wide variety of electrostatically-transduced MEMS/NEMS of practical import (e.g. gyroscopes or mode-localized sensors). Further, our work clearly describes the physics of associated intrinsic mode coupling mechanisms as opposed to previous works which simply reported the experimental observations and employed generic models to model the dynamics. A further advance is that we introduced new coupling manipulation techniques. Thus, this device is demonstrated to as a coupling-abundant multiple mode system, which is difficult to construct in a single device by a-priori engineering of the resonator.

Note that, in previous works, the only perceived nonlinear *parametric* mode coupling mechanism is due to the mechanical tension-induced nonlinearity [Physical Review Letters 105, 117205 (2010)], and most mode coupling manipulation is based on this understanding [Nature Physics 8, 387–392 (2012); Nature Physics 9, 480–484 (2013); Nano Letters 13, 1622–1626 (2013); Nano Letters 15, 6727 (2015); Nature Nanotechnology 11 (2016); Nature Nanotechnology 11, 747 (2016); etc.]

This includes previous works that employ electrostatic transduction. In this work, we demonstrate that, electrostatic field can also cause nonlinear *parametric* coupling among mechanical modes, which acts in the opposite sense to tension-induced nonlinear *parametric* coupling. Another advance is that, though electrostatic coupling has been extensively researched [Journal of Applied Physics 114, 4469 (2013); Journal of Microelectromechanical Systems 11, 802–807 (2002); *Nonlinear Dynamics of Nanomechanical and Micromechanical Resonators*, WileyVCH Verlag GmbH & Co. KGaA, 2008.], electrostatic *nonlinear* coupling among different modes in *one* resonator has never been elaborated to this level of detail.

Apart from the potential immediate practical relevance of this work to MEMS gyroscopes and other resonant sensors, this work could allow more flexible testbeds for fundamental research on mode coupling. We have revised the last paragraph of the Introduction Section and the first paragraph of the Discussion and conclusion Section to better illustrate the novelty and significance of this work.

A related point noted by the authors is the possible connection of phenomena studied here to measurable improvements in the gyro performance. For example, the authors wrote in the discussion and conclusion, “More interestingly, dynamical sideband coupling may be very useful in terms

of enhancing sensor performance...'. Could the authors follow-up on this point to show experimentally how the strong dynamical coupling can improve the gyro performance? Such a measurement that connects the exceptional control demonstrated in the paper to device performance would be of great interest to both mechanics and gyro communities. If the authors can do this, then I would support publication.

We really appreciate Reviewer #3 for giving this suggestion. In fact, improving the performance of sensor performance is one of our primary intentions and ultimate goals of this study. The application of this work is very promising.

First, the hybrid coupling and strong dynamical coupling manipulations are ideal for switching the operational modes of Coriolis vibratory gyroscopes. It is known that the in-phase bias error of the mode-matched Coriolis vibratory gyroscope will change its sign relative to the angular rate signal when drive and sense modes are switched. The most promising way to date to identify and calibrate the bias error is switching the drive and sense modes. The switching speed of the conventional stop-decay-actuate method is related to the decay time of the resonator. However, the decay time of high-performance gyroscope can be as high as dozens and even hundreds of seconds [Physical Review Applied 8, 064033 (2017); *IEEE International Conference on Micro Electro Mechanical Systems* (2019) pp. 210–213; *IEEE Sensors Letters* 1, 6500104 (2017)]. The long decay time will severely restrict the working bandwidth of the gyroscope. If we employ hybrid or dynamical coupling [Applied Physics Letters 105, 083114 (2014)] to switch the modes, the switching speed can be enhanced by orders of magnitude. Second, the electrostatic dynamical coupling has the potential to provide tunable and stable coupling for mode-localized sensors. We have implemented this approach in a mode-localised sensor with impressive improvements in sensor scale factor ($> 100x$) and resolution ($\sim 25x$) within the same device. However, this involves a significant body of work on its own and warrants a separate (more engineering-oriented) paper. Third, a deep understanding about electrostatically induced mode coupling effects of this work may provide a guideline for avoiding or utilizing mode coupling in the widely used capacitive-resonator-based sensors.

We have a long way to go in applying the techniques reported in this work to practically enhance gyroscope performance, because embedding the manipulation functions within the complex gyroscope control circuitry requires a vast amount of work. Given the novelty claim we have made above, we feel that the demonstration of gyroscope performance improvement is out of the scope of this work and should be subject of a separate manuscript. Based on the length of this paper, we hope Reviewer #3 could allow us to leave it out of the current paper and present it in a more detailed follow-up work. Demonstrating gyro performance improvement based on the coupling manipulations provided in this work is one of our most important research directions.

Based on these comments, we have enhanced the potential description in the *last paragraph of Section III DISCUSSION AND CONCLUSION*.

REVIEWERS' COMMENTS:

Reviewer #1 (Remarks to the Author):

The authors have made significant improvements over the original manuscript and I am mostly satisfied with the revision. I am glad the initial comments were helpful in improving this paper and make it more attractive and useful for readers of Nature Communications, and I am inclined to recommend it for publication.

Nevertheless, there are still improvements I urge the authors make before publication:

Fig. 2f, the hand symbols are too small and not visible. I think it is clear which mode shape refers to which, so the hand symbols are redundant. Instead, I prefer a clear downward arrow vertically pointing at both resonance peaks, especially when one of them is hardly visible. Or just put vertical dashed lines at the resonances, which is also helpful.

Page 5, line 244, 3(f) should be 3(c).

Fig. 3a and 3b, as some of the back actions are in phase and some are out of phase, the authors might consider using downward arrows for out-of-phase ones, as arrows intuitively mean vectors and arrows in the same direction intuitively refer to in-phase back actions. It would also be more helpful if the authors explain in a bit more detail about the difference of intra- and inter-modal couplings, as very little information is provided in the current version. Also—Fig 3a and 3c might give readers the misconception that one would still expect peaks at ω_{II-1} and ω_{II-2} , which are indicated by the arrows; but that's only true experimentally when $V_p=0$. So maybe the authors can come up with a way to illustrate the further splitting/avoided crossing when $V_p \neq 0$ in Fig. 3a&b, to better reflect the measurement in Fig. 3c?

Fig. 3e: making the i-v symbols more obvious is helpful. Putting them at the current location will make readers very easily miss them.

Fig. 3f: the “black dashed lines” are in fact dotted lines, and are really hard to see, more so when printed (I couldn't find them on the printed copy even when intentionally looking for them). I suggest change to the literally “dashed lines” with a heavier weight.

Page 7 line 388, “we make importance progress” .

Reviewer #2 (Remarks to the Author):

The authors have adequately addressed my concerns raised in the first round reviewing. I found the quality of the manuscript is increased by the efforts made by the authors. While reviewing the modified manuscript, I found two more minor problems which needs to be addressed before recommendation its publication in Nature Communications.

1. Fig. 4d and 4e, the authors claim that “the phenomenon is opposite to that of the mechanical nonlinear parametric coupling”. Can the authors briefly illustrate the main physical picture behind this? Why the frequency shift is opposite? Is this the major evidence that the authors claim their coupling is an electrostatic coupling rather than a mechanical coupling?

2. Some citations are inappropriate. For example, as cited in conclusion part, “some interesting dynamical sideband experiments have been implemented in capacitive mechanical systems [27, 30, 31].” Actually, Ref [27] reports tunable coupling in an architecture comprising three graphene mechanical resonators coupled in series. There is no parametric coupling or sideband measurement investigated in the paper.

Reviewer #3 (Remarks to the Author):

The revised manuscript improves the quality of illustrations and provides a detailed comparison between simulation and experiment in a microelectromechanical ring resonators. Also, the modal interaction in the MEMS resonator can be precisely controlled by the electrostatic nonlinear coupling. This controllability could allow detailed performance analysis of the gyroscope operation. Accordingly, I support publication of the paper in Nature Communication.

Once again, we express our sincerest thanks to Reviewers for their time and efforts in providing such valuable comments on this paper. We hope that the revised paper addresses concerns of the Reviewers. In the following response letter, Reviewers' comments are marked in **red with sans-serif font**, authors' responses are in **black with serif font**. The modifications are marked in **blue** in the revised manuscript.

REVIEWERS' COMMENTS:

Reviewer #1 (Remarks to the Author):

The authors have made significant improvements over the original manuscript and I am mostly satisfied with the revision. I am glad the initial comments were helpful in improving this paper and make it more attractive and useful for readers of Nature Communications, and I am inclined to recommend it for publication. Nevertheless, there are still improvements I urge the authors make before publication:

Fig. 2f, the hand symbols are too small and not visible. I think it is clear which mode shape refers to which, so the hand symbols are redundant. Instead, I prefer a clear downward arrow vertically pointing at both resonance peaks, especially when one of them is hardly visible. Or just put vertical dashed lines at the resonances, which is also helpful.

Response: This suggestion is much appreciated. We have drawn some downward arrows vertically pointing at each of the resonance peaks. Besides, we have made other revisions to remove unnecessary arrows and scale bar labels, and keep visual effects to a minimum based on the Editor's suggestion. We have tried our best to make this figure clear and address all of the suggestions above.

Page 5, line 244, 3(f) should be 3(c). Fig. 3a and 3b, as some of the back actions are in phase and some are out of phase, the authors might consider using downward arrows for out-of-phase ones, as arrows intuitively mean vectors and arrows in the same direction intuitively refer to in-phase back actions. It would also be more helpful if the authors explain in a bit more detail about the difference of intra- and inter-modal couplings, as very little information is provided in the current version.

Response: We thank Reviewer #1 for pointing out these errors, which has been corrected in the revised version of the paper. As the Reviewer suggested, we have drawn sidebands 2, 7, and 9 in downward form, because they are in antiphase with respect to drive tone, sidebands 8, and 10, respectively.

In addition, at the bottom of Page 5, we have made a statement about the difference of intra- and inter-modal coupling terms. The intra-modal coupling terms are responsible for the Stückelberg interferometry phenomenon that can be observed at the bottom of Figure 3c,d, and the inter-modal coupling term leads to the first-order dynamical coupling. Both intra- and inter-modal coupling terms are necessary for

higher-order dynamical coupling.

Also—Fig 3a and 3c might give readers the misconception that one would still expect peaks at ω_{II-1} and ω_{II-2} , which are indicated by the arrows; but that's only true experimentally when $V_p=0$. So maybe the authors can come up with a way to illustrate the further splitting/avoided crossing when $V_p < 0$ in Fig. 3a&b, to better reflect the measurement in Fig. 3c?

Response: As Reviewer #1 suggested, we have added normal-mode splitting diagrams at the bottom of Figure 3a,b, in which split resonant peaks and frequency separations (coupling rate) are illustrated.

Fig. 3e: making the i-v symbols more obvious is helpful. Putting them at the current location will make readers very easily miss them. Fig. 3f: the “black dashed lines” are in fact dotted lines, and are really hard to see, more so when printed (I couldn't find them on the printed copy even when intentionally looking for them). I suggest change to the literally “dashed lines” with a heavier weight.

Page 7 line 388, “we make importance progress”.

Response: The ‘Rainbow’ colour scale has been replaced by the ‘Reds’ colour scale in the revised figures based on the Editor's suggestion. We have made the i-v symbols much more obvious in Fig. 3e, and by choosing the background coloring of 3c and 3d. The theoretical values of mode splitting are indicated by dot-dashed lines in Fig. 3f. Thanks to the light ‘Reds’ colour scale, these dot-dash lines are much more obvious. We have corrected the grammar on Page 7, line 388.

Reviewer #2 (Remarks to the Author):

The authors have adequately addressed my concerns raised in the first round reviewing. I found the quality of the manuscript is increased by the efforts made by the authors. While reviewing the modified manuscript, I found two more minor problems which needs to be addressed before recommendation its publication in Nature Communications.

1. Fig. 4d and 4e, the authors claim that “the phenomenon is opposite to that of the mechanical nonlinear parametric coupling”. Can the authors briefly illustrate the main physical picture behind this? Why the frequency shift is opposite? Is this the major evidence that the authors claim their coupling is an electrostatic coupling rather than a mechanical coupling?

Response: We thank Reviewer #2 for making this observation. We are sorry that this explanation was not clearer in the original manuscript. In fact, the key factor for the observed mode-II frequency shift is the $3\nu_{II}x_{III}^2x_{II}$ term in the expanded equation (4) of the main file, where ν_{II} is the third-order nonlinearity coefficient. The displacement square of mode-III scaled by ν_{II} will directly influence the effective stiffness of mode-II. Thus, the frequency shift direction of mode-II is directly

determined by the sign of ν_{II} . The experimental results indicate that frequency of mode-II shifts downward, which indicates that ν_{II} is negative (stiffness-softening). Combined with the fact that this stiffness-softening nonlinearity is highly V_0 -dependent (Supplementary Note 4), we can conclude that this nonlinear parametric coupling is caused by the electrostatic field.

Further, the analysis in Supplementary Note 4, shows that the observed frequency downshifts are determined can be explained by the third order nonlinearity coefficients arising from the electrostatic field. The approximate parameters M_j and N_j ($j = 1,2$, see Supplementary equations (77-80)) that determine the frequency shifts are determined by the parameters ν_{II} , ν_{III} . An approximate analytical frequency shift model (Supplementary equations (85, 86)) is derived to and it is seen that the third-order nonlinearity coefficients ν_{II} or ν_{III} (derived by considering the nonlinear electrostatic field) determine the sign and strength of the downshift.

Based on this comment, we have made revisions to the text in the first and second paragraph of the left column of page 7 of the main file, Supplementary Equations (77-80, 85-86) and the corresponding description in page 19 of the Supplementary information.

2. Some citations are inappropriate. For example, as cited in conclusion part, “some interesting dynamical sideband experiments have been implemented in capacitive mechanical systems [27, 30, 31].” Actually, Ref [27] reports tunable coupling in an architecture comprising three graphene mechanical resonators coupled in series. There is no parametric coupling or sideband measurement investigated in the paper.

Response: We thank Reviewer #2 for pointing this out. We have removed the citation of [27] at this location and column 2 of page 1 (“capacitively transduced devices [30, 31]”). However, citation of [27] at column 1 of page 1 (“mutual coupling between two distinct mechanical resonators or modes [18–27]”) is left unchanged.

Reviewer #3 (Remarks to the Author):

The revised manuscript improves the quality of illustrations and provides a detailed comparison between simulation and experiment in a microelectromechanical ring resonators. Also, the modal interaction in the MEMS resonator can be precisely controlled by the electrostatic nonlinear coupling. This controllability could allow detailed performance analysis of the gyroscope operation. Accordingly, I support publication of the paper in Nature Communication.

Response: We thank Reviewer #3 very much for the positive comment.